# Zero-shot Generalist Graph Anomaly Detection with Unified Neighborhood Prompts

## Abstract

Graph anomaly detection (GAD), which aims to identify nodes in a graph that significantly deviate from normal patterns, plays a crucial role in broad application domains. Existing GAD methods, whether supervised or unsupervised, are one-model-for-one-dataset approaches, *i.e.*, training a separate model for each graph dataset. This limits their applicability in real-world scenarios where training on the target graph data is not possible due to issues like data privacy. To overcome this limitation, we propose a novel zero-shot generalist GAD approach **UNPrompt** that trains a one-for-all detection model, requiring the training of one GAD model on a single graph dataset and then effectively generalizing to detect anomalies in other graph datasets without any retraining or fine-tuning. The key insight in UNPrompt is that i) the predictability of latent node attributes can serve as a generalized anomaly measure and ii) highly generalized normal and abnormal graph patterns can be learned via latent node attribute prediction in a properly normalized node attribute space. UNPrompt achieves generalist GAD through two main modules: one module aligns the dimensionality and semantics of node attributes across different graphs via coordinate-wise normalization in a projected space, while another module learns generalized neighborhood prompts that support the use of latent node attribute predictability as an anomaly score across different datasets. Extensive experiments on real-world GAD datasets show that UNPrompt significantly outperforms diverse competing methods under the generalist GAD setting, and it also has strong superiority under the one-model-for-one-dataset setting.

## 1 Introduction

Graph anomaly detection (GAD) aims to identify anomalous nodes that exhibit significant deviations from the majority of nodes in a graph. GAD has attracted extensive research attention in recent years (Ma et al., 2021; Pang et al., 2021; Qiao et al., 2024) due to the board applications in various domains such as spam review detection in online shopping networks (McAuley & Leskovec, 2013; Rayana & Akoglu, 2015) and malicious user detection in social networks (Yang et al., 2019). To handle high-dimensional node attributes and complex structural relations between nodes, graph neural networks (GNNs) (Kipf & Welling, 2016; Wu et al., 2020) have been widely exploited for GAD due to their strong ability to integrate the node attributes and graph structures. These methods can be roughly divided into two categories, *i.e.*, supervised and unsupervised methods. One category formulates GAD as a binary classification problem and aims to capture anomaly patterns under the guidance of labels (Tang et al., 2022; Peng et al., 2018; Gao et al., 2023). By contrast, due to the difficulty of obtaining these class labels, another category of methods takes the unsupervised approach that aims to learn normal graph patterns, *e.g.*, via data reconstruction or other proxy learning tasks that are related to GAD (Qiao & Pang, 2023; Liu et al., 2021b; Ding et al., 2019; Huang et al., 2022).

Despite their remarkable detection performance, these methods need to train a dataset-specific model for each graph dataset for GAD. This one-model-for-one-dataset paradigm limits their applicability in real-world scenarios since training a model from scratch incurs significant computation costs and requires even a large amount of labeled data for supervised GAD methods (Liu et al., 2024; Qiao et al., 2024). Training on a target graph may even not be possible due to data privacy protection and regulation. To address this limitation, a new one-for-all anomaly detection (AD) paradigm, called generalist anomaly detection (Zhu & Pang, 2024; Zhou et al., 2024), has been proposed for image AD with the emergence of foundation models such as CLIP (Radford et al., 2021). This new

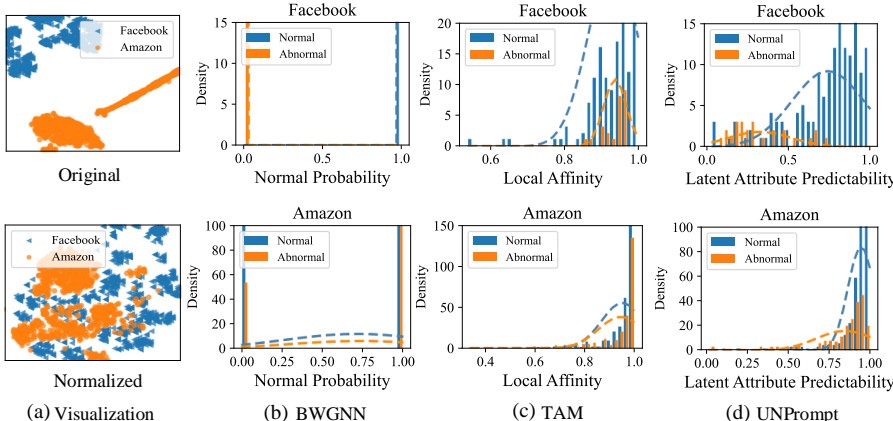

Figure 1: **(a)** Visualization of two popular GAD datasets: Facebook and Amazon, where the node attributes are unified into a common semantic space via our proposed normalization compared to the original heterogeneous raw attributes. **(b)-(d)** The anomaly scores of BWGNN (normal probability) (Tang et al., 2022), TAM (local affinity) (Qiao & Pang, 2023) and UNPrompt (latent attribute predictability) on the two datasets, where the methods are all trained on Facebook and tested on Amazon under the zero-shot setting. It is clear that BWGNN and TAM struggle to generalize from Facebook to Amazon, while UNPrompt can learn well to generalize across the datasets.

direction aims to learn a generalist detection model on auxiliary datasets so that it can generalize to detect anomalies effectively in diverse target datasets without any re-training or fine-tuning. This paper explores this direction in the area of GAD.

Compared to image AD, there are some unique challenges for learning generalist models for GAD. First, unlike image data where raw features are in the same RGB space, the node attributes in graphs from different applications and domains can differ significantly in node attribute dimensionality and semantics. For example, as a shopping network dataset, Amazon contains the relationships between users and reviews, and the node attribute dimensionality is 25. Differently, Facebook, a social network dataset, describes relationships between users with 576-dimensional attributes. Second, generalist AD models on image data rely on the superior generalizability learned in large visual-language models (VLMs) through pre-training on web-scale image-text-aligned data (Zhu & Pang, 2024; Zhou et al., 2024), whereas there are no such foundation models for graph data (Liu et al., 2023a). Therefore, the key question here is: *can we learn generalist models for GAD on graph data with heterogeneous node attributes and structure without the support of foundation models?*

To address these challenges, we propose **UNPrompt**, a novel generalist GAD approach that learns *Unified Neighborhood Prompts* on a single auxiliary graph dataset and then effectively generalizes to directly detect anomalies in other graph datasets under a **zero-shot** setting. The key insight in UNPrompt is that i) the predictability of latent node attributes can serve as a generalized anomaly measure and ii) highly generalized normal and abnormal graph patterns can be learned via latent node attribute prediction in a properly normalized node attribute space. UNPrompt achieves this through two main modules including *coordinate-wise normalization-based node attribute unification* and *neighborhood prompt learning*. The former module aligns the dimensionality of node attributes across graphs and transforms the semantics into a common space via coordinate-wise normalization, as shown in Figure 1(a). In this way, the diverse distributions of node attributes are calibrated into the same semantic space. On the other hand, the latter module learns graph-agnostic normal and abnormal patterns via a neighborhood-based latent attribute prediction task. Specifically, we incorporate learnable prompts into the normalized attributes of the neighbors of a target node to predict the latent attributes of the target node. Despite being trained on a small pre-trained GNN using a single graph, UNPrompt can effectively generalize to detect anomalous nodes in different unseen graphs without any re-training at the inference stage, as shown in Figure 1(b)-(d).

Overall, the main contributions of this paper are summarised as follows. **(1)** We propose a novel zero-shot generalist GAD approach, UNPrompt. To the best of our knowledge, this is the first method that exhibits effective zero-shot GAD performance across various graph datasets. There is a

concurrent work on generalist GAD (Liu et al., 2024), but it can only work under a few-shot setting. **(2)** We reveal that a simple yet effective coordinate-wise normalization can be utilized to unify the heterogeneous distributions in the node attributes across different graphs. **(3)** We further introduce a novel neighborhood prompt learning module that utilizes a neighborhood-based latent node attribute prediction task to learn generalized prompts in the normalized attribute space, enabling the zero-shot GAD across different graphs. **(4)** Extensive experiments on real-world GAD datasets show that UNPrompt significantly outperforms state-of-the-art competing methods under the zero-shot generalist GAD. **(5)** We show that UNPrompt can also work in the conventional one-model-for-one-dataset setting, outperforming state-of-the-art models in this popular GAD setting.

## 2 RELATED WORK

**Graph Anomaly Detection.** Existing GAD methods can be roughly categorized into unsupervised and supervised approaches (Ma et al., 2021; Qiao et al., 2024). The unsupervised methods are typically built using data reconstruction, self-supervised learning, and learnable graph anomaly measures (Qiao et al., 2024; Liu et al., 2022). The reconstruction-based approaches like DOMINANT (Ding et al., 2019) and AnomalyDAE (Fan et al., 2020) aim to capture the normal patterns in the graph, where the reconstruction error in both graph structure and attributes is utilized as the anomaly score. CoLA (Liu et al., 2021b) and SL-GAD (Zheng et al., 2021) are representative self-supervised learning methods assuming that normality is reflected in the relationship between the target node and its contextual nodes. The graph anomaly measure methods typically leverage the graph structure-aware anomaly measures to learn intrinsic normal patterns for GAD, such as node affinity in TAM (Qiao & Pang, 2023). In contrast to the unsupervised approaches, the supervised anomaly detection approaches have shown substantially better detection performance in recent years due to the incorporation of labeled anomaly data (Liu et al., 2021a; Chai et al., 2022). Most supervised methods concentrate on the design of propagation mechanisms and spectral feature transformations to address the notorious over-smoothing feature representation issues (Tang et al., 2022; Gao et al., 2023; Chai et al., 2022). Although both approaches can be adapted for zero-shot GAD by directly applying the trained GAD models to the target datasets, they struggle to capture generalized normal and abnormal patterns for GAD across different graph datasets. There are some studies working on cross-domain GAD (Ding et al., 2021b; Wang et al., 2023) that aim to transfer knowledge from a labeled graph dataset for GAD on a target dataset, but it is a fundamentally different problem from generalist GAD since cross-domain GAD requires the training on both source and target graph datasets.

**Graph Prompt Learning.** Prompt learning, initially developed in natural language processing, seeks to adapt large-scale pre-trained models to different downstream tasks by incorporating learnable prompts while keeping the pre-trained models frozen (Liu et al., 2023b). Specifically, it designs task-specific prompts capturing the knowledge of the corresponding tasks and enhances the compatibility between inputs and pre-trained models to enhance the pre-trained models in downstream tasks. Recently, prompt learning has been explored in graphs to unify multiple graph tasks (Sun et al., 2023; Liu et al., 2023c) or improve the transferability of graph models on the datasets across the different domains (Li et al., 2024; Zhao et al., 2024), *e.g.*, by optimizing the prompts with labeled data of various downstream tasks (Fang et al., 2024; Liu et al., 2023c). Although being effective in popular graph learning tasks like node classification and link prediction, they are inapplicable to generalist GAD due to the unsupervised nature and/or irregular distributions of anomalies.

**Generalist Anomaly Detection.** Generalist AD has been very recently emerging as a promising solution to tackle sample efficiency and model generalization problems in AD. There have been a few studies on non-graph data that have large pre-trained models to support the generalized pattern learning, such as image generalist AD (Zhou et al., 2023; Zhu & Pang, 2024). However, it is a very challenging task for data like graph data due to the lack of such pre-trained models. Recently a concurrent approach, ARC (Liu et al., 2024), introduces an effective framework that leverages in-context learning to achieve generalist GAD without relying on large pre-trained GNNs. Unlike ARC which focuses on a few-shot GAD setting, *i.e.*, requiring the availability of some labeled nodes in the target testing graph dataset, we tackle a zero-shot GAD setting assuming no access to any labeled data during inference stages.

**Inductive Graph Learning.** Similar to generalist setting, inductive graph learning (Hamilton et al., 2017; Ding et al., 2021a; Li et al., 2023b; Huang et al., 2023; Fang et al., 2023) also focuses

on inference on unseen graph data. However, these methods are not applicable to the generalist setting. Specifically, inductive graph learning trains the model on partial data of the whole graph dataset Hamilton et al. (2017); Ding et al. (2019); Li et al. (2023b) or the previously observed data of dynamic graphs (Fang et al., 2023). Then, the learned model is evaluated on the unseen data of the whole dataset or the future graph. These unseen testing data are from the same source of the training data with the same dimensionality and semantics. In contrast, the unseen data in our method are from different distributions/domains with significantly different dimensionality and semantics. This cross-dataset nature, specifically referred to as a zero-shot problem (Jeong et al., 2023; Zhou et al., 2024), makes our setting significantly different from the current inductive graph learning setting.

## 3 METHODOLOGY

### 3.1 PRELIMINARIES

**Notations.** Let $\mathcal{G} = (\mathcal{V}, \mathcal{E})$ be an attributed graph with $N$ nodes, where $\mathcal{V} = \{v_1, v_2, \dots, v_N\}$ represents the node set and $\mathcal{E}$ is the edge set. The attributes of nodes can be denoted as $X = [\mathbf{x}_1, \mathbf{x}_2, \dots, \mathbf{x}_N]^T \in \mathbb{R}^{N \times d}$ and the edges between nodes can be presented by an adjacency matrix $A \in \{0, 1\}^{N \times N}$ with $A_{ij} = 1$ if there is an edge between $v_i$ and $v_j$ and $A_{ij} = 0$ otherwise. For simplicity, the graph can be represented as $\mathcal{G} = (A, X)$. In GAD, the node set can be divided into a set of the normal nodes $\mathcal{V}_n$ and a set of anomalous nodes $\mathcal{V}_a$. Typically, the number of normal nodes is significantly larger than the anomalous nodes, *i.e.*, $|\mathcal{V}_n| \gg |\mathcal{V}_a|$. Moreover, the anomaly labels can be denoted as $\mathbf{y} \in \{0, 1\}^N$ with $\mathbf{y}_i = 1$ if $v_i \in \mathcal{V}_a$ and $\mathbf{y}_i = 0$ otherwise.

**Conventional GAD.** Conventional GAD typically focuses on model training and anomaly detection on the same graph. Specifically, given a graph $\mathcal{G}$, an anomaly scoring model $f : \mathcal{G} \to \mathbb{R}$ is optimized on $\mathcal{G}$ in a supervised or unsupervised manner. Then, the model is used to detect anomalies within the same graph. The model is expected to generate higher anomaly scores for abnormal nodes than normal nodes, *i.e.*, $f(v_i) < f(v_j)$ if $v_i \in \mathcal{V}_n$ and $v_j \in \mathcal{V}_a$.

**Generalist GAD.** Generalist GAD aims to learn a generalist model $f$ on a single training graph so that $f$ can be directly adapted to different target graphs across diverse domains without any fine-tuning or re-training. More specifically, the model is optimized on $\mathcal{G}_{\text{train}}$ with the corresponding anomaly labels $\mathbf{y}_{\text{train}}$. After model optimization, the learned $f$ is utilized to detect anomalies within different unseen target graphs $\mathcal{T}_{\text{test}} = \{\mathcal{G}_{\text{test}}^{(1)}, \dots, \mathcal{G}_{\text{test}}^{(n)}\}$ which has heterogeneous attributes and/or structure to $\mathcal{G}_{\text{train}}$, *i.e.*, $\mathcal{G}_{\text{train}} \cap \mathcal{T}_{\text{test}} = \emptyset$. Depending on whether labeled nodes of the target graph are provided during inference, the generalist GAD problem can be further divided into two categories, *i.e.*, **few-shot** and **zero-shot** settings. We focus on the zero-shot setting where the generalist models cannot get access to any labeled data of the testing graphs during both training and inference.

### 3.2 OVERVIEW OF THE PROPOSED APPROACH – UNPROMPT

The framework is illustrated in Figure 2, which consists of two main modules, coordinate-wise normalization-based node attribute unification and neighborhood prompt learning. For all graphs, the node attribute unification aligns the dimensionality of node attributes and transforms the semantics into a common space via coordinate-wise normalization in a projected space. Then, in the normalized space, the generalized latent attribute prediction task is performed with the neighborhood prompts to learn generalized GAD patterns at the training stage. In this prompt learning module, UNPrompt aims to maximize the predictability of the latent attributes of normal nodes while minimizing those of abnormal nodes. In this paper, we evaluate the predictability via the similarity. In doing so, the graph-agnostic normal and abnormal patterns are incorporated into the prompts. During inference, the target graph is directly fed into the learned models after node attribute unification without any re-training or labeled nodes of the graph. For each node, the predictability of latent node attributes is directly used as the normal score for final anomaly detection.

### 3.3 NODE ATTRIBUTE UNIFICATION

Graphs from different distributions and domains significantly differ in the dimensionality and semantics of node attributes. Therefore, the premise of developing a generalist GAD model is to unify

Figure 2: Overview of UNPrompt. Node attribute unification is used to align the attribute dimensionality and semantics. During training, the neighborhood prompts are optimized to capture generalized patterns by maximizing the predictability of the latent attributes (*i.e.*, the embedding $\mathbf{z}_i$) of normal nodes while minimizing that of abnormal nodes. During inference, the learned prompts are directly applied to the testing nodes, and the latent attribute predictability of each node is used for GAD.

the dimensionality and semantics of node attributes into the same space. In this paper, we propose a simple yet effective node attribute unification module to address this issue, which consists of feature projection and coordinate-wise normalization. Different from ARC (Liu et al., 2024) which aligns the attributes based on feature reordering using feature smoothness, we calibrate the feature distributions of diverse graphs into the same frame, resulting in a simpler yet effective alignment.

**Feature Projection.**  To address the inconsistent attribute dimensions across graphs, various feature projection methods can be utilized, such as singular value decomposition (Stewart, 1993) (SVD) and principal component analysis (Abdi & Williams, 2010) (PCA). Formally, given the attribute matrix $X^{(i)} \in \mathbb{R}^{N^{(i)} \times d^{(i)}}$ of any graph $\mathcal{G}^{(i)}$ from $\mathcal{G}_{\text{train}} \cup \mathcal{T}_{\text{test}}$, we transform it into $\tilde{X}^{(i)} \in \mathbb{R}^{N^{(i)} \times d'}$ with the common dimensionality of $d'$,

$$X^{(i)} \in \mathbb{R}^{N^{(i)} \times d^{(i)}} \xrightarrow[\text{Projection}]{\text{Feature}} \tilde{X}^{(i)} \in \mathbb{R}^{N^{(i)} \times d'} . \tag{1}$$

**Coordinate-wise normalization.**  Despite the attribute dimensionality being unified, the semantics and distributions of each attribute dimension are still divergent across graphs, posing significant challenges to learning a generalist GAD model. A recent study (Li et al., 2023a) has demonstrated that semantic differences across datasets are mainly reflected in the distribution shifts and calibrating the distributions into a common frame helps learn more generalized AD models. Inspired by this, we propose to use coordinate-wise normalization to align the semantics and unify the distributions across graphs. Specifically, the transformed attribute matrix $\tilde{X}^{(i)}$ is shifted and re-scaled to have mean zeros and variance ones via the following equation:

$$\bar{X}^{(i)} = \frac{\tilde{X}^{(i)} - \boldsymbol{\mu}^{(i)}}{\boldsymbol{\sigma}^{(i)}} , \tag{2}$$

where $\boldsymbol{\mu}^{(i)} = [\mu_1^{(i)}, \ldots, \mu_{d'}^{(i)}]$ and $\boldsymbol{\sigma}^{(i)} = [\sigma_1^{(i)}, \ldots, \sigma_{d'}^{(i)}]$ are the coordinate-wise mean and variance of $\tilde{X}^{(i)}$ of the graph $\mathcal{G}^{(i)}$. In this way, the distributions of normalized attributes along each dimension are the same within and across graphs, as shown in Figure 1(a). This helps to capture the generalized normal and abnormal patterns for generalist GAD (see Table 2).

## 3.4 NEIGHBORHOOD PROMPT LEARNING VIA LATENT NODE ATTRIBUTE PREDICTION

**Latent Node Attribute Predictability as Anomaly Score.**  To build a generalist GAD model, one must capture the generalized normal and abnormal patterns across graphs. Otherwise, the model would overfit the dataset-specific knowledge of the training graph which can be very different from that in target graphs. In this paper, we reveal that the predictability of latent node attributes can serve as a generalized anomaly measure, and thus, highly generalized normal and abnormal graph

patterns can be learned via latent node attribute prediction in the normalized node attribute space with the neighborhood prompts. The key intuition of this anomaly measure is that normal nodes tend to have more connections with normal nodes of similar attributes due to prevalent graph homophily relations, resulting in a more homogeneous neighborhood in the normal nodes (Qiao & Pang, 2023); by contrast, the presence of anomalous connections and/or attributes makes abnormal nodes deviate significantly from their neighbors. Therefore, for a target node, its latent attributes (*i.e.*, node embedding) is more predictable based on the latent attributes of its neighbors if the node is normal node, compared to abnormal nodes. The neighborhood-based latent attribute prediction is thus used to measure the normality for GAD. As shown in our experiments (see Figures 1(b)-(d) and Tables 1 and 3), it is a generalized anomaly scoring method that works effectively across graphs. However, due to the existence of irrelevant and noisy attribute information in the original attribute space, the attribute prediction is not as effective as expected in the simply projected space after attribute unification. To address this issue, we propose to learn discriminative prompts via the latent attribute prediction task to enhance the effectiveness of this anomaly measure.

To achieve this, we first design a simple graph neural network $g$, a neighborhood aggregation network, to generate the aggregated neighborhood embedding of each target node. Specifically, given a graph $\mathcal{G} = (A, \bar{X})$, the aggregated neighborhood embeddings for each node are obtained as follows:

$$\tilde{Z} = g(\mathcal{G}) = \tilde{A}\bar{X}W \,, \tag{3}$$

where $\tilde{Z}$ is the aggregated representation of neighbors, $\tilde{A} = (D)^{-1}A$ is the normalized adjacency matrix with $D$ being a diagonal matrix and its elements $D_{kk} = \sum_j A_{kj}$, and $W$ is the learnable parameters. Compared to conventional GNNs such as GCN (Kipf & Welling, 2016) and SGC (Wu et al., 2019), we do not require $\tilde{A}$ to be self-looped and symmetrically normalized as we aim to obtain the aggregated representation of all the neighbors for each node. To design the latent node attribute prediction task, we further obtain the latent attributes of each node as follows:

$$Z = \bar{X}W \,, \tag{4}$$

where $Z$ serves as the prediction ground truth for the latent attribute prediction task. The adjacency matrix $A$ is discarded to avoid carrying neighborhood-based attribute information into $Z$ which would lead to ground truth leakage in this prediction task. We further propose to utilize the cosine similarity to measure this neighborhood-based latent attribute predictability for each node:

$$s_i = \text{sim}(\mathbf{z}_i, \tilde{\mathbf{z}}_i) = \frac{\mathbf{z}_i(\tilde{\mathbf{z}}_i)^T}{\|\mathbf{z}_i\|\|\tilde{\mathbf{z}}_i\|} \,, \tag{5}$$

where $\mathbf{z}$ and $\tilde{\mathbf{z}}_i$ are the $i$-th node embeddings in $Z$ and $\tilde{Z}$ respectively. A higher similarity denotes the target node can be well predicted by its neighbors and indicates the target is normal with a higher probability. Therefore, we directly utilize the similarity to measure the normal score of the nodes.

**GNN Pre-training.** To build generalist models, pre-training is required. Here we pre-train the above neighborhood aggregation network via graph contrastive learning due to the ability to obtain robust and transferable models (You et al., 2020; Zhu et al., 2020) across graphs (see Appendix B for the details). Without pre-training, the dataset-specific knowledge would be captured by the model if it is directly optimized based on the neighborhood-based latent attribute prediction of normal and abnormal nodes, limiting the generalizability of the model to other graphs (see Table 2).

**Neighborhood Prompting via Latent Attribute Prediction.** After the pre-training, we aim to further learn more generalized normal and abnormal patterns via prompt tuning in the normalized space. Thus, we devise learnable prompts appending to the attributes of the neighboring nodes of the target nodes, namely *neighborhood prompts*, for learning robust and discriminative patterns that can detect anomalous nodes in different unseen graphs without any re-training during inference.

Specifically, neighborhood prompting aims to learn some prompt tokens that help maximize the neighborhood-based latent prediction of normal nodes while minimizing that of abnormal nodes simultaneously. To this end, the prompt is designed as a set of shared and learnable tokens that can be incorporated into the normalized node attributes. Formally, the neighborhood prompts are represented as $P = [\mathbf{p}_1, \ldots, \mathbf{p}_k]^T \in \mathbb{R}^{K \times d'}$ where $K$ is the number of vector-based tokens $\mathbf{p}_i$. For each

node in $\mathcal{G} = (A, \bar{X})$, the node attributes in the unified feature space are augmented by the weighted combination of these tokens, with the weights obtained from $K$ learnable linear projections:

$$\hat{\mathbf{x}}_i = \bar{\mathbf{x}}_i + \sum_j^K \alpha_j \mathbf{p}_j\,, \quad \alpha_j = \frac{e^{(\mathbf{w}_j)^T \mathbf{x}_i^t}}{\sum_l^K e^{(\mathbf{w}_l)^T \mathbf{x}_i^t}}\,, \tag{6}$$

where $\alpha_j$ denotes the importance score of the token $\mathbf{p}_j$ in the prompt and $\mathbf{w}_j$ is a learnable projection. For convenience, we denote the graph modified by the graph prompt as $\tilde{\mathcal{G}} = (A, \bar{X} + P)$. Then, $\tilde{\mathcal{G}}$ is fed into the frozen pre-trained model $g$ to obtain the corresponding aggregated embeddings $\tilde{Z}$ and node latent attributes $Z$ via Eq.(3) and Eq.(4) respectively to measure the attribute predictability. To further enhance the representation discrimination, a transformation layer $h$ is applied on the learned $\tilde{Z}$ and $Z$ to transform them into a more anomaly-discriminative feature space,

$$\tilde{Z} = h(\tilde{Z})\,, \quad Z = h(Z)\,. \tag{7}$$

The transformed representations are then used to measure the latent node attribute predictability with Eq.(5). To optimize $P$ and $h$, we employ the following training objective,

$$\min_{P,h} \ \sum \ell(\mathbf{z}_i, \tilde{\mathbf{z}}_i)\,, \tag{8}$$

where $\ell(\mathbf{z}_i, \tilde{\mathbf{z}}_i) = -\mathrm{sim}(\mathbf{z}_i, \tilde{\mathbf{z}}_i)$ if $\mathbf{y}_i = 0$, and $\ell(\mathbf{z}_i, \tilde{\mathbf{z}}_i) = \mathrm{sim}(\mathbf{z}_i, \tilde{\mathbf{z}}_i)$ if $\mathbf{y}_i = 1$.

### 3.5 Training and Inference of UNPrompt

**Training.** The training process of UNPrompt can be divided into two parts. First, given $\mathcal{G}_{\text{train}}$, a neighborhood aggregation network $g$ is optimized via graph contrastive learning. Then, the neighborhood prompts $P$ and the transformation layer $h$ are optimized to capture the graph-agnostic normal and abnormal patterns while keeping the pre-trained model $g$ frozen. In this way, the transferable knowledge of the pre-trained $g$ is maintained, while the neighborhood prompt learning helps learn the generalized normal and abnormal patterns.

**Inference.** During inference, given $\mathcal{G}_{\text{test}}^{(i)} \in \mathcal{T}_{\text{test}}$, the node attributes are first aligned. Then, the test graph $\mathcal{G}_{\text{test}}^{(i)}$ is augmented with the learned neighborhood prompt $P$ and fed into the model $g$ and the transformation layer $h$ to obtain the neighborhood aggregated representations and the latent node attributes. Finally, the similarity (Eq.(5)) is used as the normal score for the test nodes for anomaly detection. Note that the inference does not require any further re-training and labeled nodes of $\mathcal{G}_{\text{test}}^{(i)}$. The algorithms of the training and inference of UNPrompt are provided in Appendix C.

## 4 Experiments

### 4.1 Performance on Zero-shot Generalist GAD

**Datasets.** We evaluate the proposed UNPrompt on seven real-world GAD datasets from diverse social networks and online shopping co-review networks. Specifically, the social networks include Facebook (Xu et al., 2022), Reddit (Kumar et al., 2019) and Weibo (Kumar et al., 2019). The co-review networks consist of Amazon (McAuley & Leskovec, 2013), YelpChi (Rayana & Akoglu, 2015), Amazon-all (McAuley & Leskovec, 2013) and YelpChi-all (Rayana & Akoglu, 2015).

**Competing Methods.** Since there is no zero-shot generalist GAD method, a set of eight state-of-the-art (SotA) unsupervised and supervised competing methods are employed for comparison in our experiments. The unsupervised methods comprise reconstruction-based AnomalyDAE (Fan et al., 2020), contrastive learning-based CoLA (Liu et al., 2021b), hop prediction-based HCM-A (Huang et al., 2022), local affinity-based TAM (Qiao & Pang, 2023) and GADAM (Chen et al., 2024). Supervised methods include two conventional GNNs – GCN (Kipf & Welling, 2016) and GAT (Veličković et al., 2017) – and three SotA GAD GNNs – BWGNN (Tang et al., 2022), GHRN (Gao et al., 2023) and XGBGraph (Tang et al., 2023).

Following (Liu et al., 2024; Qiao & Pang, 2023; Qiao et al., 2024), two widely-used metrics, AU-ROC and AUPRC, are used to evaluate the performance of all methods. For both metrics, the higher value denotes the better performance. Moreover, for each method, we report the average performance with standard deviations after 5 independent runs with different random seeds.

Table 1: AUROC and AUPRC results on six real-world GAD datasets with the models trained on Facebook only. For each dataset, the best performance per column within each metric is boldfaced, with the second-best underlined. "Avg" denotes the averaged performance of each method.

| Metric | Method | Dataset | | | | | | |
|---|---|---|---|---|---|---|---|---|
| | | Amazon | Reddit | Weibo | YelpChi | Aamzon-all | YelpChi-all | Avg. |
| AUROC | *Unsupervised Methods* | | | | | | | |
| | AnomalyDAE | $0.5818_{\pm 0.039}$ | $0.5016_{\pm 0.032}$ | $\underline{0.7785}_{\pm 0.058}$ | $0.4837_{\pm 0.094}$ | $0.7228_{\pm 0.023}$ | $0.5002_{\pm 0.018}$ | $\underline{0.5948}$ |
| | CoLA | $0.4580_{\pm 0.054}$ | $0.4623_{\pm 0.005}$ | $0.3924_{\pm 0.041}$ | $0.4907_{\pm 0.017}$ | $0.4091_{\pm 0.052}$ | $0.4879_{\pm 0.010}$ | $0.4501$ |
| | HCM-A | $0.4784_{\pm 0.005}$ | $0.5387_{\pm 0.041}$ | $0.5782_{\pm 0.048}$ | $0.5000_{\pm 0.000}$ | $0.5056_{\pm 0.059}$ | $0.5023_{\pm 0.005}$ | $0.5172$ |
| | GADAM | $\underline{0.6646}_{\pm 0.063}$ | $0.4532_{\pm 0.024}$ | $0.3652_{\pm 0.052}$ | $0.3376_{\pm 0.012}$ | $0.5959_{\pm 0.080}$ | $\underline{0.4829}_{\pm 0.016}$ | $0.4832$ |
| | TAM | $0.4720_{\pm 0.005}$ | $\mathbf{0.5725}_{\pm 0.004}$ | $0.4867_{\pm 0.028}$ | $0.5035_{\pm 0.014}$ | $0.7543_{\pm 0.002}$ | $0.4216_{\pm 0.002}$ | $0.5351$ |
| | *Supervised Methods* | | | | | | | |
| | GCN | $0.5988_{\pm 0.016}$ | $\underline{0.5645}_{\pm 0.000}$ | $0.2232_{\pm 0.074}$ | $0.5366_{\pm 0.019}$ | $0.7195_{\pm 0.002}$ | $\underline{0.5486}_{\pm 0.001}$ | $0.5319$ |
| | GAT | $0.4981_{\pm 0.008}$ | $0.5000_{\pm 0.025}$ | $0.4521_{\pm 0.101}$ | $\underline{0.5871}_{\pm 0.016}$ | $0.5005_{\pm 0.012}$ | $0.4802_{\pm 0.004}$ | $0.5030$ |
| | BWGNN | $0.4769_{\pm 0.020}$ | $0.5208_{\pm 0.016}$ | $0.4815_{\pm 0.108}$ | $0.5538_{\pm 0.027}$ | $0.3648_{\pm 0.050}$ | $0.5282_{\pm 0.015}$ | $0.4877$ |
| | GHRN | $0.4560_{\pm 0.033}$ | $0.5253_{\pm 0.006}$ | $0.5318_{\pm 0.038}$ | $0.5524_{\pm 0.020}$ | $0.3382_{\pm 0.085}$ | $0.5125_{\pm 0.016}$ | $0.4860$ |
| | XGBGraph | $0.4179_{\pm 0.000}$ | $0.4601_{\pm 0.000}$ | $0.5373_{\pm 0.000}$ | $0.5722_{\pm 0.000}$ | $\underline{0.7950}_{\pm 0.000}$ | $0.4945_{\pm 0.000}$ | $0.5462$ |
| | UNPrompt (Ours) | $\mathbf{0.7525}_{\pm 0.016}$ | $0.5337_{\pm 0.002}$ | $\mathbf{0.8860}_{\pm 0.007}$ | $\mathbf{0.5875}_{\pm 0.016}$ | $\mathbf{0.7962}_{\pm 0.022}$ | $\mathbf{0.5558}_{\pm 0.012}$ | $\mathbf{0.6853}$ |
| AUPRC | *Unsupervised Methods* | | | | | | | |
| | AnomalyDAE | $0.0833_{\pm 0.015}$ | $0.0327_{\pm 0.004}$ | $\underline{0.6064}_{\pm 0.031}$ | $0.0624_{\pm 0.017}$ | $0.1921_{\pm 0.026}$ | $0.1484_{\pm 0.009}$ | $\underline{0.1876}$ |
| | CoLA | $0.0669_{\pm 0.002}$ | $0.0391_{\pm 0.004}$ | $0.1189_{\pm 0.014}$ | $0.0511_{\pm 0.000}$ | $0.0861_{\pm 0.019}$ | $0.1466_{\pm 0.003}$ | $0.0848$ |
| | HCM-A | $0.0669_{\pm 0.002}$ | $0.0391_{\pm 0.004}$ | $0.1189_{\pm 0.014}$ | $0.0511_{\pm 0.000}$ | $0.0861_{\pm 0.019}$ | $0.1466_{\pm 0.003}$ | $0.0848$ |
| | GADAM | $\underline{0.1562}_{\pm 0.103}$ | $0.0293_{\pm 0.001}$ | $0.0830_{\pm 0.005}$ | $0.0352_{\pm 0.001}$ | $0.1595_{\pm 0.121}$ | $0.1371_{\pm 0.006}$ | $0.1001$ |
| | TAM | $0.0666_{\pm 0.001}$ | $\underline{0.0413}_{\pm 0.001}$ | $0.1240_{\pm 0.014}$ | $0.0524_{\pm 0.002}$ | $0.1736_{\pm 0.004}$ | $0.1240_{\pm 0.001}$ | $0.0970$ |
| | *Supervised Methods* | | | | | | | |
| | GCN | $\underline{0.0891}_{\pm 0.007}$ | $\mathbf{0.0439}_{\pm 0.000}$ | $0.1109_{\pm 0.020}$ | $0.0648_{\pm 0.009}$ | $0.1536_{\pm 0.002}$ | $\underline{0.1735}_{\pm 0.000}$ | $0.1060$ |
| | GAT | $0.0688_{\pm 0.002}$ | $0.0329_{\pm 0.002}$ | $0.1009_{\pm 0.017}$ | $\mathbf{0.0810}_{\pm 0.005}$ | $0.0696_{\pm 0.001}$ | $0.1400_{\pm 0.002}$ | $0.0822$ |
| | BWGNN | $0.0652_{\pm 0.002}$ | $0.0389_{\pm 0.003}$ | $0.2241_{\pm 0.046}$ | $0.0708_{\pm 0.018}$ | $0.0586_{\pm 0.003}$ | $0.1605_{\pm 0.005}$ | $0.1030$ |
| | GHRN | $0.0633_{\pm 0.003}$ | $0.0407_{\pm 0.002}$ | $0.1965_{\pm 0.059}$ | $0.0661_{\pm 0.010}$ | $0.0569_{\pm 0.006}$ | $0.1505_{\pm 0.005}$ | $0.0957$ |
| | XGBGraph | $0.0536_{\pm 0.000}$ | $0.0330_{\pm 0.000}$ | $0.2256_{\pm 0.000}$ | $0.0655_{\pm 0.000}$ | $\underline{0.2307}_{\pm 0.000}$ | $0.1449_{\pm 0.000}$ | $0.1256$ |
| | UNPrompt (Ours) | $\mathbf{0.1602}_{\pm 0.013}$ | $0.0351_{\pm 0.000}$ | $\mathbf{0.6406}_{\pm 0.026}$ | $\underline{0.0712}_{\pm 0.008}$ | $\mathbf{0.2430}_{\pm 0.028}$ | $\mathbf{0.1810}_{\pm 0.012}$ | $\mathbf{0.2219}$ |

**Implementation Details.** To ensure a fair comparison, the common dimensionality is set to eight to unify the node attribute across graphs for all methods, and SVD is used for feature projection. The number of layers in GNNs is set to one and the number of hidden units is 128. The transformation layer is also implemented via a one-layer MLP with the same number of hidden units. The size of the neighborhood prompt is set to one by default. For all baselines, we adopt their official code and follow the recommended optimization and hyperparameter settings to conduct the experiments. UNPrompt and all its competing methods are trained on Facebook and then directly tested on the other six GAD datasets without any further training or additional knowledge of the target graphs.

**Main Results.** The AUROC and AUPRC results of all methods are presented in Table 1. From the table, we can have the following observations. (1) Under the proposed generalist GAD scenario where a model is trained on a single dataset and evaluated on six other datasets, all the competing baselines fail to work well, demonstrating that it is very challenging to build a generalist GAD model that generalizes across different datasets under zero-shot setting. (2) For supervised methods, the simple GCN achieves better performance than the specially designed GAD GNNs. This can be attributed to more dataset-specific knowledge being captured in these specialized GAD models, limiting their generalization capacity to the unseen testing graphs. (3) Unsupervised methods perform more stable than supervised methods across the target graphs and generally outperform supervised ones. This is because the unsupervised objectives are closer to the shared anomaly patterns across graphs compared to the supervised ones, especially for TAM which employs a fairly generalized local affinity-based objective to train the model. (4) The proposed method UNPrompt demonstrates strong and stable generalist GAD capacity across graphs from different distributions and domains. Specifically, UNPrompt achieves the best AUROC performance on 5 out of 6 datasets and the average performance outperforms the best-competing method by over 9%. In terms of AUPRC, UNPrompt outperforms all baselines on 4 out of 6 datasets and also achieves the best average performance. The superiority of UNPrompt is attributed to the fact that i) the proposed coordinate-wise normalization effectively aligns the features across graphs, and ii) the shared generalized normal and abnormal patterns are well captured in the neighborhood prompts.

**Ablation Study.** To evaluate the importance of each component in UNPrompt, we design four variants, *i.e.*, w/o coordinate-wise normalization, w/o graph contrastive learning-based pre-training, without neighborhood prompts, and w/o transformation layer. The results of these variants are re-

Table 2: AUROC results of the proposed method UNPrompt and its four variants.

| Method | Amazon | Reddit | Weibo | YelpChi | Aamzon-all | YelpChi-all | Avg. |
|---|---|---|---|---|---|---|---|
| UNPrompt | **0.7525** | 0.5337 | **0.8860** | **0.5875** | **0.7962** | **0.5558** | **0.6853** |
| w/o Normalization | 0.4684 | 0.5006 | 0.1889 | 0.5620 | 0.3993 | 0.5466 | 0.4443 |
| w/o Pre-training | 0.5400 | 0.5233 | 0.5658 | 0.4672 | 0.3902 | 0.4943 | 0.4968 |
| w/o Prompt | 0.5328 | 0.5500 | 0.4000 | 0.4520 | 0.4096 | 0.4894 | 0.4723 |
| w/o Transformation | 0.7331 | **0.5556** | 0.7406 | 0.5712 | 0.7691 | 0.5545 | 0.6540 |

ported in the Table 2. From the table, we can see that all four components contribute to the overall superior performance of UNPrompt. More specifically, (1) without the coordinate-wise normalization, the method fails to calibrate the distributions of diverse node attributes into a common space, leading to large performance drop across all datasets. (2) Besides the semantics alignment, the graph contrastive learning-based pre-training ensures our GNN network is transferable to other graphs instead of overfitting to the training graph. As expected, the performance of the variant without pre-training also drops significantly. (3) If the neighborhood prompts are removed, the learning of latent node attribute prediction is ineffective for capturing generalized normal and abnormal patterns. (4) The variant without the transformation layer achieves inferior performance on nearly all the datasets, demonstrating the importance of mapping the features into a more anomaly-discriminative space.

**Sensitivity w.r.t the Neighborhood Prompt Size.** We evaluate the sensitivity of UNPrompt w.r.t the size of the neighborhood prompts, *i.e.*, the number of tokens $K$. We vary $K$ in the range of $[1, 9]$ and report the results in Figure 3(a). It is clear that the performances on Reddit, Weibo and YelpChi-all remain stable with varying sizes of neighborhood prompts while the other datasets show slight fluctuation, demonstrating that the generalized normal and abnormal patterns can be effectively captured in our neighborhood prompts even with a small size.

**Prompt learning using latent attribute prediction vs. alternative graph anomaly measures.** To further justify the effectiveness of latent attribute predictability on learning generalized GAD patterns in our prompt learning module, we compare this proposed learnable anomaly measure to the recently proposed anomaly measure, local node affinity in TAM (Qiao & Pang, 2023). All modules of UNPrompt are fixed with only the latent attribute prediction task replaced as the maximization of local affinity as in TAM. The results are presented in Figure 3(b). We can see that the latent attribute predictability consistently and significantly outperforms the local affinity-based measure across all graphs, demonstrating its superiority in learning generalized patterns for generalist GAD.

## 4.2 Performance on Conventional Unsupervised GAD

We also evaluate the applicability of UNPrompt unsupervised GAD setting to further verify the effectiveness of the latent note attribute prediction-based anomaly scores using our proposed neighborhood prompt learning. Specifically, we adopt the same pipeline as in the generalist GAD setting, *i.e.*, graph contrastive-based pre-training and neighborhood prompt learning. Different from the training process in the generalist setting, there is no label information available in unsupervised GAD since models are trained and evaluated on the same graph data. To address this issue, we employ the pseudo-labeling technique to provide supervision for neighborhood prompt learning. In a nutshell, we enforce the neighborhood prompts to maximize the latent attribute predictability of high-score nodes. More details on unsupervised GAD are provided in Appendix D.

**Experimental Setup.** Six datasets from different distributions and domains are used, *i.e.*, Amazon, Facebook, Reddit, YelpCHi, Amazon-all, and YelpChi-all. Following (Qiao & Pang, 2023), eight SotA unsupervised baselines are used for comparison, *i.e.*, iForest (Liu et al., 2012), ANOMALOUS (Peng et al., 2018), CoLA (Liu et al., 2021b), SL-GAD (Zheng et al., 2021), HCM-A (Huang et al., 2022), DOMINANT (Ding et al., 2019), ComGA (Luo et al., 2022) and TAM (Qiao & Pang, 2023). For each method, we report the average performance with standard deviations after 5 independent runs with different random seeds. The implementation details of UNPrompt remain the same as in the generalist GAD setting. More experimental details on unsupervised GAD are in Appendix F.2.

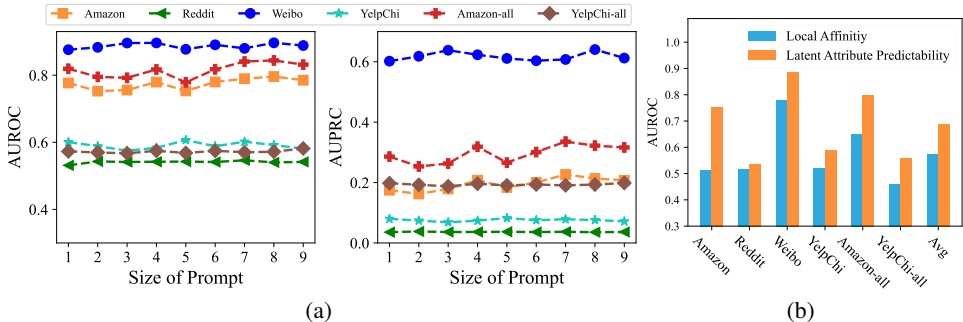

Figure 3: (**a**) AUROC and AUPRC results of UNPrompt w.r.t. varying neighborhood prompt size. (**b**). The AUROC performance of generalist GAD with different prompt learning objectives.

Table 3: AUROC and AUPRC results of unsupervised GAD methods on six real-world GAD datasets. The best performance per column within each metric is boldfaced, with the second-best underlined. "Avg" denotes the averaged performance of each method.

| Metric | Method | Dataset | | | | | | Avg. |
|---|---|---|---|---|---|---|---|---|
| | | Amazon | Facebook | Reddit | YelpChi | Amazon-all | YelpChi-all | |
| AUROC | iForest | $0.5621_{\pm 0.008}$ | $0.5382_{\pm 0.015}$ | $0.4363_{\pm 0.020}$ | $0.4120_{\pm 0.040}$ | $0.1914_{\pm 0.002}$ | $0.3617_{\pm 0.001}$ | 0.4169 |
| | ANOMALOUS | $0.4457_{\pm 0.003}$ | $0.9021_{\pm 0.005}$ | $0.5387_{\pm 0.012}$ | $0.4956_{\pm 0.003}$ | $0.3230_{\pm 0.021}$ | $0.3474_{\pm 0.018}$ | 0.5087 |
| | DOMINANT | $0.5996_{\pm 0.004}$ | $0.5677_{\pm 0.002}$ | $0.5555_{\pm 0.011}$ | $0.4133_{\pm 0.010}$ | $0.6937_{\pm 0.028}$ | $0.5390_{\pm 0.014}$ | 0.5615 |
| | CoLA | $0.5898_{\pm 0.008}$ | $0.8434_{\pm 0.011}$ | $\underline{0.6028}_{\pm 0.007}$ | $0.4636_{\pm 0.001}$ | $0.2614_{\pm 0.021}$ | $0.4801_{\pm 0.016}$ | 0.5402 |
| | SL-GAD | $0.5937_{\pm 0.011}$ | $0.7936_{\pm 0.005}$ | $0.5677_{\pm 0.005}$ | $0.3312_{\pm 0.035}$ | $0.2728_{\pm 0.012}$ | $0.5551_{\pm 0.015}$ | 0.5190 |
| | HCM-A | $0.3956_{\pm 0.014}$ | $0.7387_{\pm 0.032}$ | $0.4593_{\pm 0.011}$ | $0.4593_{\pm 0.005}$ | $0.4191_{\pm 0.011}$ | $0.5691_{\pm 0.018}$ | 0.5069 |
| | ComGA | $0.5895_{\pm 0.008}$ | $0.6055_{\pm 0.000}$ | $0.5453_{\pm 0.003}$ | $0.4391_{\pm 0.000}$ | $0.7154_{\pm 0.014}$ | $0.5352_{\pm 0.006}$ | 0.5716 |
| | TAM | $\underline{0.7064}_{\pm 0.010}$ | $\underline{0.9144}_{\pm 0.008}$ | $0.6023_{\pm 0.004}$ | $\underline{0.5643}_{\pm 0.007}$ | $\underline{0.8476}_{\pm 0.028}$ | $\underline{0.5818}_{\pm 0.033}$ | $\underline{0.7028}$ |
| | UNPrompt (Ours) | $\mathbf{0.7335}_{\pm 0.020}$ | $\mathbf{0.9379}_{\pm 0.006}$ | $\mathbf{0.6067}_{\pm 0.006}$ | $\mathbf{0.6223}_{\pm 0.007}$ | $\mathbf{0.8516}_{\pm 0.004}$ | $\mathbf{0.6084}_{\pm 0.001}$ | $\mathbf{0.7267}$ |
| AUPRC | iForest | $0.1371_{\pm 0.002}$ | $0.0316_{\pm 0.003}$ | $0.0269_{\pm 0.001}$ | $0.0409_{\pm 0.000}$ | $0.0399_{\pm 0.001}$ | $0.1092_{\pm 0.001}$ | 0.0643 |
| | ANOMALOUS | $0.0558_{\pm 0.001}$ | $0.1898_{\pm 0.004}$ | $0.0375_{\pm 0.004}$ | $0.0519_{\pm 0.002}$ | $0.0321_{\pm 0.001}$ | $0.0361_{\pm 0.005}$ | 0.0672 |
| | DOMINANT | $0.1424_{\pm 0.002}$ | $0.0314_{\pm 0.041}$ | $0.0356_{\pm 0.002}$ | $0.0395_{\pm 0.002}$ | $0.1015_{\pm 0.018}$ | $0.1638_{\pm 0.007}$ | 0.0857 |
| | CoLA | $0.0677_{\pm 0.001}$ | $0.2106_{\pm 0.017}$ | $\underline{0.0449}_{\pm 0.002}$ | $0.0448_{\pm 0.002}$ | $0.0516_{\pm 0.001}$ | $0.1361_{\pm 0.015}$ | 0.0926 |
| | SL-GAD | $0.0634_{\pm 0.005}$ | $0.1316_{\pm 0.020}$ | $0.0406_{\pm 0.004}$ | $0.0350_{\pm 0.000}$ | $0.0444_{\pm 0.001}$ | $0.1711_{\pm 0.011}$ | 0.0810 |
| | HCM-A | $0.0527_{\pm 0.015}$ | $0.0713_{\pm 0.004}$ | $0.0287_{\pm 0.005}$ | $0.0287_{\pm 0.012}$ | $0.0565_{\pm 0.003}$ | $0.1154_{\pm 0.004}$ | 0.0589 |
| | ComGA | $0.1153_{\pm 0.005}$ | $0.0354_{\pm 0.001}$ | $0.0374_{\pm 0.001}$ | $0.0423_{\pm 0.000}$ | $0.1854_{\pm 0.003}$ | $0.1658_{\pm 0.003}$ | 0.0969 |
| | TAM | $\underline{0.2634}_{\pm 0.008}$ | $\underline{0.2233}_{\pm 0.016}$ | $0.0446_{\pm 0.001}$ | $\underline{0.0778}_{\pm 0.009}$ | $\underline{0.4346}_{\pm 0.021}$ | $\underline{0.1886}_{\pm 0.017}$ | $\underline{0.2054}$ |
| | UNPrompt (Ours) | $\mathbf{0.2688}_{\pm 0.060}$ | $\mathbf{0.2622}_{\pm 0.028}$ | $\mathbf{0.0450}_{\pm 0.001}$ | $\mathbf{0.0895}_{\pm 0.004}$ | $\mathbf{0.6094}_{\pm 0.014}$ | $\mathbf{0.2068}_{\pm 0.004}$ | $\mathbf{0.2470}$ |

**Main Results.** The AUROC and AUPRC results of all methods are presented in Table 3. Despite being a generalist GAD method, UNPrompt works very well as a specialized GAD model too. UNPrompt substantially outperforms all the competing methods on all datasets in terms of both AUROC and AUPRC. Particularly, the average performance of UNPrompt surpasses the best-competing method TAM by over 2% in both metrics. Moreover, UNPrompt outperforms the best-competing method by 2%-6% in AUROC on most of the datasets. The superior performance shows that the latent node attribute predictability can be a generalized GAD measure that holds for different graphs, and this property can be effectively learned by the proposed neighborhood prompting method.

## 5 CONCLUSION

In this paper, we propose a novel zero-shot generalist GAD method, UNPrompt, that trains one detector on a single dataset and can effectively generalize to other unseen target graphs without any further re-training or labeled nodes of target graphs during inference. The attribute inconsistency and the absence of generalized anomaly patterns are the main obstacles for generalist GAD. To address these issues, two main modules are proposed, *i.e.*, coordinate-wise normalization-based attribute unification and neighborhood prompt learning. The first module aligns node attribute dimensionality and semantics, while the second module captures generalized normal and abnormal patterns via the neighborhood-based latent node attribute prediction. Extensive experiments on various real-world GAD datasets from different distributions and domains demonstrate the effectiveness of UNPrompt for generalist GAD. Besides, the experiments conducted on the unsupervised GAD with UNPrompt further support the rationality of the learned anomaly patterns in the generalist model.

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

## A  GRAPH SIMILARITY

In addition to the visualization results presented in Figure 1, we further provide the distributional similarity of various graphs in this section. Specifically, for dimension-aligned graphs across different distributions and domains, we measure their distributional similarity to analyze their diverse semantics.

Given a graph $\mathcal{G}^{(i)} = (A^{(i)}, \tilde{X}^{(i)})$, the coordinate-wise mean $\boldsymbol{\mu}^{(i)} = [\mu_1^{(i)}, \ldots, \mu_{d'}^{(i)}]$ and variance $\boldsymbol{\sigma}^{(i)} = [\sigma_1^{(i)}, \ldots, \sigma_{d'}^{(i)}]$ of $\tilde{X}^{(i)}$ are calculated and concatenated to form the distributional vector of $\mathcal{G}^{(i)}$, *i.e.*, $\mathbf{d}_i = [\boldsymbol{\mu}^{(i)}, \boldsymbol{\sigma}^{(i)}]$. Then, the distribution similarity between $\mathcal{G}^{(i)}$ and $\mathcal{G}^{(j)}$ is measured via the cosine similarity,

$$s_{ij} = \mathrm{sim}(\mathbf{d}_i, \mathbf{d}_j). \tag{9}$$

The distributional similarities between graphs from different domains or distributions are shown in Figure 4(a).. From the figure, we can see that the distributional similarities are typically small, demonstrating the diverse semantics of node features across graphs. Noth that, for Amazon & Amazon-all and YelpChi & YelpChi-all, their distribution similarity is one, which can be attributed to the fact that they are from the same distributions respectively but with different numbers of nodes and structures.

To reduce the semantic gap among graphs for generalist GAD, we propose to calibrate the distributions of all graphs into the same frame with coordinate-wise normalization. The distributional similarity with normalization is illustrated in Figure 4(b). It is clear that the node attributes share the same distribution after the normalization. In this way, the generalist model can better capture the shared GAD patterns and generalize to different target graphs, as demonstrated in our experimental results.

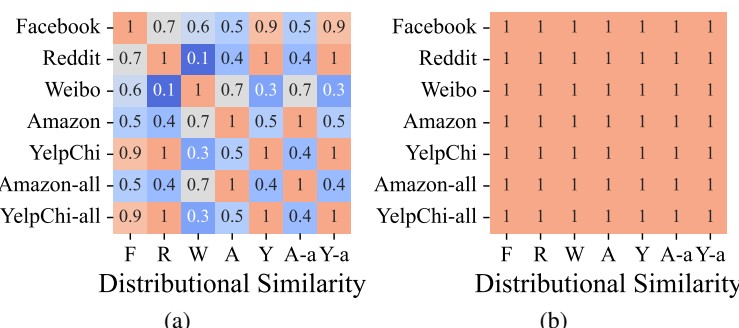

Figure 4: **(a)** Distributional similarity between different graphs without coordinate-wise normalization. **(b)** Distributional similarity between different graphs with the coordinate-wise normalization.

## B  DETAILS ON PRE-TRAINING OF NEIGHBORHOOD AGGREGATION NETWORKS

We pre-train the neighborhood aggregation network $g$ via graph contrastive learning (Zhu et al., 2020) for subsequent graph prompt learning so that the generic normality and abnormality can be captured in the prompts.

Specifically, given the training graph $\mathcal{G} = (A, X)$, to construct contrastive views for graph contrastive learning, two widely used graph augmentations are employed, *i.e.*, edge removal and attribute masking (Zhu et al., 2020). The edge removal randomly drops a certain portion of existing edges in $\mathcal{G}$ and the attribute masking randomly masks a fraction of dimensions with zeros in node attributes, *i.e.*,

$$\hat{A} = A \circ R, \quad \hat{X} = [\mathbf{x}_1 \circ \mathbf{m}, \ldots, \mathbf{x}_N \circ \mathbf{m}]^T, \tag{10}$$

where $R \in \{0, 1\}^{N \times N}$ is the edge masking matrix whose entry is drawn from a Bernoulli distribution controlled by the edge removal probability, $\mathbf{m} \in \{0, 1\}^d$ is the attribute masking vector whose

entry is independently drawn from a Bernoulli distribution with the attribute masking ratio, and $\circ$ denotes the Hadamard product.

By applying the graph augmentations to the original graph, the corrupted graph $\hat{\mathcal{G}} = (\hat{A}, \hat{X})$ forms the contrastive view for the original graph $\mathcal{G} = (A, X)$. Then, $\hat{\mathcal{G}}$ and $\mathcal{G}$ are fed to the shared model $g$ followed by the non-linear projection to obtain the corresponding node embeddings, $i.e.$, $\hat{Z}'$ and $Z'$. For graph contrastive learning, the embeddings of the same node in different views are pulled closer while the embeddings of other nodes are pushed apart. The pairwise objective for each node pair $(\hat{\mathbf{z}}'_i, \mathbf{z}'_i)$ can be formulated as:

$$\ell(\hat{\mathbf{z}}'_i, \mathbf{z}'_i) = -\log \frac{e^{\text{sim}(\hat{\mathbf{z}}'_i, \mathbf{z}'_i)/\tau}}{e^{\text{sim}(\hat{\mathbf{z}}'_i, \mathbf{z}'_i)/\tau} + \sum_{j\neq i}^{N} e^{\text{sim}(\hat{\mathbf{z}}'_i, \mathbf{z}'_j)/\tau} + \sum_{j\neq i}^{N} e^{\text{sim}(\hat{\mathbf{z}}'_i, \hat{\mathbf{z}}'_j)/\tau}} \,, \tag{11}$$

where $\text{sim}(\cdot)$ represents the cosine similarity and $\tau$ is a temperature hyperparameter. Therefore, the overall objective can be defined as follows:

$$\mathcal{L}_{\text{contrast}} = \frac{1}{2N} \sum_{i=1}^{N} (\ell(\hat{\mathbf{z}}'_i, \mathbf{z}'_i) + \ell(\mathbf{z}'_i, \hat{\mathbf{z}}'_i)) \,. \tag{12}$$

With the objective Eq.(12), the model $g$ is optimized to learn transferable discriminative representations of nodes.

## C  ALGORITHMS

The training and inference processes of UNPrompt are summarized in Algorithms 1 and Algorithm 2, respectively.

---

**Algorithm 1:** Training of UNPrompt

---

1: **Input:** Training graph $\mathcal{G}_{\text{train}} = (A, X)$; training epoch $E$
2: **Output:** Neighborhood aggregation network $g$, graph prompts $P = [\mathbf{p}_1, \ldots, \mathbf{p}_K]$, and transformation $h$.
3: Perform feature unification of $X$.
4: Pre-train $g$ on $\mathcal{G}_{\text{train}}$ with graph contrastive learning in Eq.( 12).
5: Keep model $g$ frozen.
6: **for** $e = 1, \ldots, E$ **do**
7:    Obtain modified node attribute with prompts via Eq.(6).
8:    Obtain the neighborhood aggregated representation $\tilde{Z}$ via Eq.(3).
9:    Obtain the node representations $Z$ via Eq.(4).
10:    Transform $\tilde{Z}$ and $Z$ with $h$ via Eq.(7).
11:    Optimize $P$ and $h$ by minimizing Eq.(8).
12: **end for**

---

---

**Algorithm 2:** Inference of UNPrompt

---

1: **Input:** Testing graphs $\mathcal{T}_{\text{test}} = \{\mathcal{G}_{\text{test}}^{(1)}, \ldots, \mathcal{G}_{\text{test}}^{(n)}\}$, neighborhood aggregation network $g$, graph prompts $P = [\mathbf{p}_1, \ldots, \mathbf{p}_K]$, and transformation $h$.
2: **Output:** Normal score of testing nodes.
3: **for** $\mathcal{G}_{\text{test}}^{(i)} = (A^{(i)}, X^{(i)}) \in \mathcal{T}_{\text{test}}$ **do**
4:    Perform feature unification of $X^{(i)}$.
5:    Obtain modified node attribute with prompts via Eq.(6).
6:    Obtain the neighborhood aggregated representation $\tilde{Z}^{(i)}$ via Eq.(3).
7:    Obtain the node representations $Z^{(i)}$ via Eq.(4).
8:    Transform $\tilde{Z}^{(i)}$ and $Z^{(i)}$ with $h$ via Eq.(7).
9:    Obtain the normal score of nodes via Eq.(5).
10: **end for**

---

## D  UNSUPERVISED GAD WITH UNPROMPT

To demonstrate the wide applicability of the proposed method UNPrompt, we further perform unsupervised GAD with UNPrompt which focuses on detecting anomalous nodes within one graph and does not have access to any node labels during training. Specifically, we adopt the same pipeline in the generalist GAD setting, *i.e.*, graph contrastive pertaining and neighborhood prompt learning. Since we focus on anomaly detection on each graph separately, the node attribute unification is discarded for unsupervised GAD. However, the absence of node labels poses a challenge to learning meaningful neighborhood prompts for anomaly detection. To overcome this issue, we propose to utilize the pseudo-labeling technique to guide the prompt learning. Specifically, the normal score of each node is calculated by the neighborhood-based latent attribute predictability after the graph contrastive learning process:

$$s_i = \text{sim}(\mathbf{z}_i, \tilde{\mathbf{z}}_i),  \tag{13}$$

where $\mathbf{z}_i$ is the node representation learned by graph contrastive learning and $\tilde{\mathbf{z}}_i$ is the corresponding aggregated neighborhood representation. Higher $s_i$ of node $v_i$ typically indicates a higher probability of $v_i$ being normal nodes. Therefore, more emphasis should be put on high-score nodes when learning neighborhood prompts. To achieve this, the normal score $s_i$ is transformed into the loss weight $w_i = \text{Sigmoid}(\alpha(s_i - t))$ where $t$ is a threshold and $\alpha$ is the scaling parameter. In this way, $w_i$ would approach 1 if $s_i > t$ and 0 otherwise. Overall, the objective for unsupervised GAD using UNPrompt can be formulated as follows:

$$\mathcal{L} = \sum_i^N (-w_i \text{sim}(\mathbf{z}_i, \tilde{\mathbf{z}}_i) + \lambda \sum_{j,j\neq i}^N \text{sim}(\mathbf{z}_i, \tilde{\mathbf{z}}_j)),  \tag{14}$$

where the second term is a regularization term employed to prevent the node embeddings from being collapsed into the same and $\lambda$ is a trade-off hyperparameter.

Note that, we only focus on maximizing the latent attribute predictability of high-score nodes without minimizing the predictability of low-score nodes in the above objective. These low-score nodes could also be normal nodes with high probability as the score from Eq.(13) is only obtained from the pre-trained model, resulting in the score not being fully reliable. If the predictability is also minimized for these nodes, conflicts would be induced for neighborhood prompt learning, limiting the performance of unsupervised GAD. After optimization, the latent attribute predictability is also directly used as the normal score for the final unsupervised GAD.

## E  TIME COMPLEXITY ANALYSIS

**Theoretical Analysis.** In this section, we analyze the time complexity of training UNPrompt. As discussed in the main paper, UNPrompt first pre-trains the aggregation network with graph contrastive learning. Then, the model remains frozen when optimizing neighborhood graph prompts and the transformation layer to capture the generalized normal and abnormal graph patterns. In the experimental section, we employ a one-layer aggregation network, denoting the number of hidden units as $d_h$. The time complexity of the graph contrastive learning is $\mathcal{O}(4E_1(|A|d_h + Nd_h d' + 6Nd_h^2))$, where $|A|$ returns of the number of edges of the $\mathcal{G}_{\text{train}}$, $N$ is the number of nodes, $d'$ represents the predefined dimensionality of node attributes, and $E_1$ is the number of training epoch. After that, we freeze the learned model and learn the learnable neighborhood prompt tokens and the transformation layer to capture the shared anomaly patterns. In our experiments, we set the size of each graph prompt to $K$ and implement the classification head as a single-layer MLP with the same hidden units $d_h$. Given the number of the training epoch $E_2$, the time complexity of optimizing the graph prompt and the transformation layer is $\mathcal{O}((4KNd' + 2Nd_h^2)E_2)$, which includes both the forward and backward propagation. Note that, despite the neighborhood aggregation model being frozen, the forward and backward propagations of the model are still needed to optimize the task-specific graph prompts and the transformation layer. Therefore, the overall time complexity of UNPrompt is $\mathcal{O}(4E_1(|A|d_h + Nd_h d' + 6Nd_h^2) + 2E_2(|A|d_h + Nd_h d' + 2KNd' + Nd_h^2))$, which is linear to the number of nodes, the number of edges, and the number of node attributes of the training graph. Note that, after the training, the learned generalist model is directly utilized to perform anomaly detection on various target graphs without any further training.

Table 4: Training time and inference time (seconds) for different methods.

| Methods | AnomalyDAE | TAM | GAT | BWGNN | UNPrompt (Ours) |
|---|---|---|---|---|---|
| Training Time | 86.04 | 479.70 | 2.43 | 4.86 | 2.08 |
| Inference Time | 264.29 | 521.92 | 300.90 | 330.99 | 58.95 |

**Empirical Computational Complexity Analysis.** In Table 4, we report the training time and inference time of different methods, where two representative unsupervised methods (AnomalyDAE and TAM) and two supervised methods (GAT and BWGNN) are used for comparison to our method UNPrompt. The results show that the proposed method requires much less training and inference time compared to other baselines, demonstrating the efficiency of the proposed UNPrompt. Note that, TAM has the highest time consumption, which can be attributed to that it performs multiple graph truncation and learns multiple local affinity maximization networks.

## F EXPERIMENTAL SETUP

### F.1 DETAILS ON DATASETS

We conduct the experiments using seven real-world with genuine anomalies in diverse online shopping services and social networks, including Facebook (Xu et al., 2022), Reddit (Kumar et al., 2019), Weibo (Zhao et al., 2020), Amazon (McAuley & Leskovec, 2013) and YelpChi (Rayana & Akoglu, 2015) as well as two large-scale graph datasets including Amazon-all (McAuley & Leskovec, 2013) and YelpChi-all (Rayana & Akoglu, 2015). The statistical information including the number of nodes, edge, the dimension of the feature, and the anomaly rate of the datasets can be found in Table 5. The more detailed description of each dataset is given as follows

- **Facebook** (Xu et al., 2022). Facebook is a social network where each node represents a user, and edges signify relationships between users. Ground truth anomalies are are nodes that either connect to randomly selected circles or exhibit abnormal attributes, as described in (Ding et al., 2019; Liu et al., 2021b).

- **Reddit** (Kumar et al., 2019). Reddit is a forum-based network derived from the social media platform Reddit, where nodes represent users, and the embeddings of post texts serve as attributes. Users who have been banned from the platform are labeled as anomalies.

- **Weibo** (Kumar et al., 2019). Weibo is a social network and their associated hashtags are obtained from the Tencent Weibo platform. Users who engaged in at least five of these activities are labeled as anomalies while the others are classified as normal samples. Suspicious activities are defined as two posts made within a specific timeframe, such as 60 seconds. The attributes of each node include the location of a micro-blog post and bag-of-words features.

- **Amazon** (McAuley & Leskovec, 2013). Amazon is a graph dataset that captures the relations between users and product reviews. There are 25 handcrafted features used as the node attribute (Zhang et al., 2020). The users with more than 80% helpful votes are labeled as normal entities and users with less than 20% helpful votes as anomalies. Amazon is constructed by extracting the Amazon-UPU dataset that connects the users who give reviews to at least one common product.

- **YelpChi** (Rayana & Akoglu, 2015). YelpChi includes hotel and restaurant reviews filtered (spam) and recommended (legitimate) by Yelp. There are 32 handcrafted features used as node attributes (Rayana & Akoglu, 2015). The users with more than 80% helpful votes are labeled as benign entities and users with less than 20% helpful votes as fraudulent entities. The YelpChi is constructed by extracting YelpChi-RUR which connects reviews posted by the same user.

- **Amazon-all** (McAuley & Leskovec, 2013). Amazon-all includes three types of relations: U-P-U (users reviewing at least one same product), U-S-U (users giving at least one same star rating within one week), and U-V-U (users with top-5% mutual review similarities). Amazon-all is formed by treating the different relations as a single relation following Chen et al. (2022); Qiao & Pang (2023).

Table 5: Key statistics of the real-world GAD datasets with real anomalies.

| Data set | Type | Nodes | Edges | Attributes | Anomalies(Rate) |
|---|---|---|---|---|---|
| Facebook | Social Networks | 1,081 | 55,104 | 576 | 27(2.49%) |
| Reddit | Social Networks | 10,984 | 168,016 | 64 | 366(3.33%) |
| Weibo | Social Networks | 8,405 | 407,963 | 400 | 868(10.30%) |
| Amazon | Co-review | 10,244 | 175,608 | 25 | 693(6.66%) |
| YelpChi | Co-review | 24,741 | 49,315 | 32 | 1,217(4.91%) |
| Amazon-all | Co-review | 11,944 | 4,398,392 | 25 | 821(6.87%) |
| YelpChi-all | Co-review | 45,941 | 3,846,979 | 32 | 6,674(14.52%) |
| Disney | Co-purchase | 124 | 335 | 28 | 6(4.84%) |
| Elliptic | Payment Flow | 203,769 | 234,355 | 166 | 4,545(9.76%) |

- **YelpChi-all** (Rayana & Akoglu, 2015). Similar to Amazon-all, YelpChi-all includes three types of edges: R-U-R (reviews posted by the same user), R-S-R (reviews for the same product with the same star rating), and R-T-R (reviews for the same product posted in the same month). YelpChi-all is formed by treating the different relations as a single relation following Chen et al. (2022); Qiao & Pang (2023) .

## F.2 MORE IMPLEMENTATION DETAILS

**Generalist GAD.** For the graph contrastive learning-based pre-training, the probabilities of edge removal and attribute masking are by default set to 0.2 and 0.3 respectively. Besides, the learning rate is set to 0.001 with the Adam optimizer, the training epoch is set to 200 and the temperature $\tau$ is 0.5.

For the neighborhood prompt learning, the learning rate is also set to 0.001 with the Adam optimizer, and the training epoch is set to 900. Note that, since we focus on generalist GAD, we do not perform any hyperparameter search for specific target graphs. Instead, the results of all target graphs are obtained with the same hyperparameter settings.

**Unsupervised GAD.** Similar to the generalist GAD setting, the hidden units of the neighborhood aggregation network and the transformation layer are set to 128 for all graphs. The threshold $t$ is determined by the 40th percentile of the normal scores obtained by the pre-trained model $g$, and the scaling parameter $\alpha$ is set to 10 for all graphs. Besides, we utilize random search to find the optimal hyperparameters of the size of neighborhood prompts $K$ and the trade-off parameter $\lambda$.

For both generalist and unsupervised GAD, the code is implemented with Pytorch (version: 1.13.1), DGL (version: 1.0.1), OGB (version: 1.3.6), and Python 3.8.19. All experiments are conducted on a Linux server with an Intel CPU (Intel Xeon Gold 6346 3.1GHz) and a Nvidia A40 GPU.

## G MORE EXPERIMENTAL RESULTS

### G.1 GENERALIST PERFORMANCE WITH DIFFERENT COMMON DIMENSIONALITIES

For the results reported in the main paper, the common dimensionality is set to eight. In this subsection, we further evaluate the generalist anomaly detection with different common dimensionalities. Specifically, the dimensionality varies in $[2, 4, 6, 8, 10, 12]$ and the results are reported in Figure 5.

From the figure, we can see that small dimensionality leads to poor generalist anomaly detection performance. This is attributed to the fact that much attribute information would be discarded with a small dimensionality. By increasing the common dimensionality, more attribute information is retained, generally resulting in much better detection performance.

### G.2 RESULTS ON TWO OTHER GRAPHS FROM DIFFERENT DOMAINS

Besides the social networks and co-review graphs, we further evaluate the performance of UN-Prompt on Disney (Liu et al., 2022) and Elliptic (Weber et al., 2019). These two datasets consist of

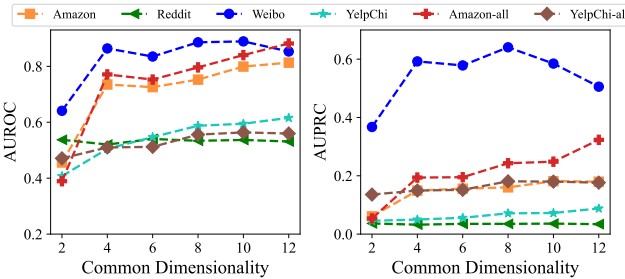

Figure 5: AUROC and AUPRC results of UNPrompt w.r.t. varying common dimensionality.

Table 6: AUROC and AUPRC results on two additional real-world GAD datasets with the models trained on Facebook only. For each dataset, the best performance per column within each metric is boldfaced, with the second-best underlined. "Avg" denotes the averaged performance of each method.

| Metric | Method | Dataset | | Avg. |
|---|---|---|---|---|
| | | Disney | Elliptic | |
| AUROC | Unsupervised Methods | | | |
| | AnomalyDAE | $0.4853_{\pm 0.003}$ | $0.4197_{\pm 0.109}$ | 0.4525 |
| | CoLA | $0.4696_{\pm 0.065}$ | $0.5572_{\pm 0.019}$ | 0.5134 |
| | HCM-A | $0.2014_{\pm 0.015}$ | $0.2975_{\pm 0.004}$ | 0.2495 |
| | GADAM | $0.4288_{\pm 0.023}$ | $0.3922_{\pm 0.012}$ | 0.4105 |
| | TAM | $0.4773_{\pm 0.003}$ | $0.3282_{\pm 0.003}$ | 0.4028 |
| | Supervised Methods | | | |
| | GCN | $0.5000_{\pm 0.000}$ | $\mathbf{0.7640}_{\pm 0.002}$ | **0.6320** |
| | GAT | $0.5175_{\pm 0.054}$ | $\underline{0.6588}_{\pm 0.019}$ | 0.5882 |
| | BWGNN | $0.6073_{\pm 0.026}$ | $0.5843_{\pm 0.101}$ | 0.5958 |
| | GHRN | $0.5336_{\pm 0.030}$ | $0.5400_{\pm 0.103}$ | 0.5368 |
| | XGBGraph | $\mathbf{0.6692}_{\pm 0.000}$ | $0.4274_{\pm 0.000}$ | 0.5483 |
| | UNPrompt (Ours) | $\underline{0.6412}_{\pm 0.030}$ | $0.5901_{\pm 0.026}$ | $\underline{0.6157}$ |
| AUPRC | Unsupervised Methods | | | |
| | AnomalyDAE | $0.0566_{\pm 0.000}$ | $0.0798_{\pm 0.014}$ | 0.0682 |
| | CoLA | $0.0701_{\pm 0.023}$ | $0.0998_{\pm 0.005}$ | 0.0850 |
| | HCM-A | $0.0355_{\pm 0.001}$ | $0.0776_{\pm 0.000}$ | 0.0566 |
| | GADAM | $0.0651_{\pm 0.012}$ | $0.0733_{\pm 0.001}$ | 0.0692 |
| | TAM | $0.0628_{\pm 0.001}$ | $0.0697_{\pm 0.001}$ | 0.0663 |
| | Supervised Methods | | | |
| | GCN | $0.0484_{\pm 0.000}$ | $\mathbf{0.1963}_{\pm 0.002}$ | $\underline{0.1224}$ |
| | GAT | $0.0530_{\pm 0.004}$ | $\underline{0.1366}_{\pm 0.010}$ | 0.0948 |
| | BWGNN | $0.0624_{\pm 0.003}$ | $0.1158_{\pm 0.026}$ | 0.0891 |
| | GHRN | $0.0519_{\pm 0.003}$ | $0.1148_{\pm 0.041}$ | 0.0834 |
| | XGBGraph | $\underline{0.1215}_{\pm 0.000}$ | $0.0816_{\pm 0.000}$ | 0.1016 |
| | UNPrompt (Ours) | $\mathbf{0.1236}_{\pm 0.031}$ | $0.1278_{\pm 0.004}$ | **0.1257** |

co-purchase network and financial network respectively. The statistics of them are also summarized in Table 5. We follow exactly the same experimental settings in the main paper.

The results of all competing methods are reported in Table 6. It is clear that UNPrompt can still achieve promising performance, demonstrating the generality of UNPrompt across different graphs. Although GCN and XGBGraph obtain the best AUROC performance on Disney and Elliptic respectively, they perform poorly on most of the other datasets. UNPrompt ranks in second in the average AUROC performance and the best AUPRC performance here. This is consisentent with the superior performance of UNPrompt in Table 1.

### G.3 INCORPORATING COORDINATE-WISE NORMALIZATION INTO BASELINES

We further conduct experiments by incorporating the proposed coordinate-wise normalization into the baselines to evaluate whether the normalization could facilitate the baselines. Without loss of the generality, three unsupervised methods (AnomalyDAE, CoLA and TAM) and three supervised methods (GCN, BWGNN and GHRN) are used and the results are reported in Table 7.

From the table, we can see that the proposed coordinate-wise normalization does not improve the baselines consistently but downgrades most of the baselines. This can be attributed to two reasons. First, while the proposed coordinate-wise normalization unifies the semantics of different graphs into the common space, the discrimination between normal and abnormal patterns would also be

Table 7: AUROC and AUPRC results of several baselines with coordinate-wise normalization (CN).

| Metric | Method | Dataset | | | | | | |
| --- | --- | --- | --- | --- | --- | --- | --- | --- |
| | | Amazon | Reddit | Weibo | YelpChi | Aamzon-all | YelpChi-all | Avg. |
| | | *Unsupervised Methods* | | | | | | |
| | AnomalyDAE | $0.5818_{\pm 0.039}$ | $0.5016_{\pm 0.032}$ | $\underline{0.7785}_{\pm 0.058}$ | $0.4837_{\pm 0.094}$ | $0.7228_{\pm 0.023}$ | $0.5002_{\pm 0.018}$ | $\underline{0.5948}$ |
| | + CN | $0.4359_{\pm 0.053}$ | $0.4858_{\pm 0.063}$ | $0.4526_{\pm 0.074}$ | $\mathbf{0.5992}_{\pm 0.028}$ | $0.2833_{\pm 0.039}$ | $0.5080_{\pm 0.013}$ | $0.4608$ |
| | CoLA | $0.4580_{\pm 0.054}$ | $0.4623_{\pm 0.005}$ | $0.3924_{\pm 0.041}$ | $0.4907_{\pm 0.017}$ | $0.4091_{\pm 0.052}$ | $0.4879_{\pm 0.010}$ | $0.4501$ |
| | + CN | $0.4729_{\pm 0.019}$ | $0.5299_{\pm 0.008}$ | $0.3401_{\pm 0.026}$ | $0.3640_{\pm 0.006}$ | $0.5424_{\pm 0.019}$ | $0.4882_{\pm 0.008}$ | $0.4563$ |
| | TAM | $0.4720_{\pm 0.005}$ | $\mathbf{0.5725}_{\pm 0.004}$ | $0.4867_{\pm 0.028}$ | $0.5035_{\pm 0.014}$ | $0.7543_{\pm 0.002}$ | $0.4216_{\pm 0.002}$ | $0.5351$ |
| | + CN | $0.4509_{\pm 0.015}$ | $0.5526_{\pm 0.006}$ | $0.4723_{\pm 0.007}$ | $0.5189_{\pm 0.006}$ | $\underline{0.7580}_{\pm 0.004}$ | $0.4057_{\pm 0.002}$ | $0.5264$ |
| AUROC | | *Supervised Methods* | | | | | | |
| | GCN | $\underline{0.5988}_{\pm 0.016}$ | $\underline{0.5645}_{\pm 0.000}$ | $0.2232_{\pm 0.074}$ | $0.5366_{\pm 0.019}$ | $0.7195_{\pm 0.002}$ | $\underline{0.5486}_{\pm 0.001}$ | $0.5319$ |
| | + CN | $0.5694_{\pm 0.014}$ | $0.5349_{\pm 0.008}$ | $0.0632_{\pm 0.005}$ | $0.3954_{\pm 0.002}2$ | $0.6798_{\pm 0.009}$ | $0.5550_{\pm 0.005}$ | $0.4663$ |
| | BWGNN | $0.4769_{\pm 0.020}$ | $0.5208_{\pm 0.016}$ | $0.4815_{\pm 0.108}$ | $0.5538_{\pm 0.027}$ | $0.3648_{\pm 0.050}$ | $0.5282_{\pm 0.015}$ | $0.4877$ |
| | + CN | $0.4745_{\pm 0.048}$ | $0.4942_{\pm 0.011}$ | $0.2538_{\pm 0.038}$ | $0.4727_{\pm 0.016}$ | $0.6307_{\pm 0.077}$ | $0.5221_{\pm 0.025}$ | $0.4747$ |
| | GHRN | $0.4560_{\pm 0.033}$ | $0.5253_{\pm 0.006}$ | $0.5318_{\pm 0.038}$ | $0.5524_{\pm 0.020}$ | $0.3382_{\pm 0.085}$ | $0.5125_{\pm 0.016}$ | $0.4860$ |
| | + CN | $0.4308_{\pm 0.024}$ | $0.5061_{\pm 0.026}$ | $0.2621_{\pm 0.043}$ | $0.4781_{\pm 0.018}$ | $0.5712_{\pm 0.046}$ | $0.5200_{\pm 0.009}$ | $0.4614$ |
| | UNPrompt (Ours) | $\mathbf{0.7525}_{\pm 0.016}$ | $0.5337_{\pm 0.002}$ | $\mathbf{0.8860}_{\pm 0.007}$ | $\underline{0.5875}_{\pm 0.016}$ | $\mathbf{0.7962}_{\pm 0.022}$ | $\mathbf{0.5558}_{\pm 0.012}$ | $\mathbf{0.6853}$ |
| | | *Unsupervised Methods* | | | | | | |
| | AnomalyDAE | $0.0833_{\pm 0.015}$ | $0.0327_{\pm 0.004}$ | $\underline{0.6064}_{\pm 0.031}$ | $0.0624_{\pm 0.017}$ | $0.1921_{\pm 0.026}$ | $0.1484_{\pm 0.009}$ | $\underline{0.1876}$ |
| | + CN | $0.0596_{\pm 0.009}$ | $0.0333_{\pm 0.007}$ | $0.1910_{\pm 0.049}$ | $\mathbf{0.0874}_{\pm 0.011}$ | $0.0495_{\pm 0.006}$ | $0.1527_{\pm 0.007}$ | $0.0956$ |
| | CoLA | $0.0669_{\pm 0.002}$ | $0.0391_{\pm 0.004}$ | $0.1189_{\pm 0.014}$ | $0.0511_{\pm 0.000}$ | $0.0861_{\pm 0.019}$ | $0.1466_{\pm 0.003}$ | $0.0848$ |
| | + CN | $0.0669_{\pm 0.002}$ | $0.0360_{\pm 0.002}$ | $0.1618_{\pm 0.027}$ | $0.0370_{\pm 0.000}$ | $0.0934_{\pm 0.002}$ | $0.1446_{\pm 0.005}$ | $0.0899$ |
| | TAM | $0.0666_{\pm 0.001}$ | $\underline{0.0413}_{\pm 0.001}$ | $0.1240_{\pm 0.014}$ | $0.0524_{\pm 0.002}$ | $0.1736_{\pm 0.004}$ | $0.1240_{\pm 0.001}$ | $0.0970$ |
| | + CN | $0.0606_{\pm 0.003}$ | $0.0394_{\pm 0.001}$ | $0.1044_{\pm 0.005}$ | $0.0542_{\pm 0.001}$ | $\mathbf{0.2482}_{\pm 0.013}$ | $0.1213_{\pm 0.001}$ | $0.1047$ |
| AUPRC | | *Supervised Methods* | | | | | | |
| | GCN | $\underline{0.0891}_{\pm 0.007}$ | $\mathbf{0.0439}_{\pm 0.000}$ | $0.1109_{\pm 0.020}$ | $0.0648_{\pm 0.009}$ | $0.1536_{\pm 0.002}$ | $\underline{0.1735}_{\pm 0.000}$ | $0.1060$ |
| | + CN | $0.0770_{\pm 0.003}$ | $0.0355_{\pm 0.001}$ | $0.0548_{\pm 0.000}$ | $0.0401_{\pm 0.000}$ | $0.1383_{\pm 0.006}$ | $0.1789_{\pm 0.002}$ | $0.0874$ |
| | BWGNN | $0.0652_{\pm 0.002}$ | $0.0389_{\pm 0.003}$ | $0.2241_{\pm 0.046}$ | $0.0708_{\pm 0.018}$ | $0.0586_{\pm 0.003}$ | $0.1605_{\pm 0.005}$ | $0.1030$ |
| | + CN | $0.0684_{\pm 0.014}$ | $0.0320_{\pm 0.001}$ | $0.2576_{\pm 0.031}$ | $0.0516_{\pm 0.004}$ | $0.1557_{\pm 0.115}$ | $0.1585_{\pm 0.010}$ | $0.1206$ |
| | GHRN | $0.0633_{\pm 0.009}$ | $0.0407_{\pm 0.002}$ | $0.1965_{\pm 0.059}$ | $0.0661_{\pm 0.010}$ | $0.0569_{\pm 0.006}$ | $0.1505_{\pm 0.005}$ | $0.0957$ |
| | + CN | $0.0586_{\pm 0.004}$ | $0.0330_{\pm 0.002}$ | $0.2663_{\pm 0.038}$ | $0.0525_{\pm 0.004}$ | $0.0898_{\pm 0.015}$ | $0.1570_{\pm 0.007}$ | $0.1095$ |
| | UNPrompt (Ours) | $\mathbf{0.1602}_{\pm 0.013}$ | $0.0351_{\pm 0.000}$ | $\mathbf{0.6406}_{\pm 0.026}$ | $\underline{0.0712}_{\pm 0.008}$ | $\underline{0.2430}_{\pm 0.028}$ | $\mathbf{0.1810}_{\pm 0.012}$ | $\mathbf{0.2219}$ |

compressed. This requires the generalist anomaly detector to capture the fine-grained differences between normal and abnormal patterns. Second, these baselines are not designed to capture generalized abnormality and normality across graphs, failing to capture and discriminate the generalized nuance. By contrast, we reveal that the predictability of latent node attributes can serve as a generalized anomaly measure and learn highly generalized normal and abnormal patterns via latent node attribute prediction. In this way, the graph-agnostic anomaly measure can be well generalized across graphs.

## G.4 GENERALIST PERFORMANCE WITH DIFFERENT TRAINING GRAPH

In the main paper, we report the generalist performance of UNPrompt by using Facebook as the training graph. To further demonstrate the generalizability of UNPrompt, we conduct additional experiments by using Amazon as the training graph and testing the learned generalist model on the rest graphs. Note that, Facebook and Amazon are from different domains, which are the social network and co-review network respectively.

The AUROC and AUPRC results of all methods are reported in Table 8. Similar to the observations when taking Facebook as the training graph, UNPrompt achieves the best average performance in terms of both AUROC and AUPRC when training on Amazon, demonstrating the generalizability and effectiveness of UNPrompt with different training graphs. Note that, the training graph Amazon and the target graph Amazon-all come from the same distribution but have different numbers of nodes and graph structures. Intuitively, all the methods should achieve promising performance on Amazon-all. However, only a few methods achieve this goal, including BWGNN, GHRN, and our method. The failures of other baselines can be attributed to the more complex graph structure of Amazon-all hinders the generalizability of these methods. Moreover, compared to BWGNN and GHRN, our method performs more stably across different datasets. This demonstrates the importance of capturing intrinsic normal and abnormal patterns for graph anomaly detection.

Table 8: AUROC and AUPRC results on six real-world GAD datasets with the generalist model trained on Amazon. For each dataset and metric, the best performance per column is boldfaced, with the second-best underlined. "Avg" denotes the averaged performance of each method.

| Metric | Method | Dataset | | | | | | Avg. |
|---|---|---|---|---|---|---|---|---|
| | | Facebook | Reddit | Weibo | YelpChi | Aamzon-all | YelpChi-all | |
| AUROC | | Unsupervised Methods | | | | | | |
| | AnomalyDAE | $0.6123_{\pm0.141}$ | $\mathbf{0.5799}_{\pm0.035}$ | $\underline{0.7884}_{\pm0.031}$ | $0.4788_{\pm0.046}$ | $0.6233_{\pm0.070}$ | $0.4912_{\pm0.009}$ | $0.5957$ |
| | CoLA | $0.5427_{\pm0.109}$ | $0.4962_{\pm0.025}$ | $0.3987_{\pm0.017}$ | $0.3358_{\pm0.012}$ | $0.4751_{\pm0.014}$ | $0.4937_{\pm0.003}$ | $0.4570$ |
| | HCM-A | $0.5044_{\pm0.047}$ | $0.4993_{\pm0.057}$ | $0.4937_{\pm0.056}$ | $0.5000_{\pm0.000}$ | $0.4785_{\pm0.016}$ | $0.4958_{\pm0.003}$ | $0.4953$ |
| | GADAM | $0.6024_{\pm0.033}$ | $0.4720_{\pm0.062}$ | $0.4324_{\pm0.047}$ | $0.4299_{\pm0.023}$ | $0.5199_{\pm0.072}$ | $0.5289_{\pm0.017}$ | $0.4976$ |
| | TAM | $0.5496_{\pm0.038}$ | $\underline{0.5764}_{\pm0.003}$ | $0.4876_{\pm0.029}$ | $0.5091_{\pm0.014}$ | $0.7525_{\pm0.002}$ | $0.4268_{\pm0.002}$ | $0.5503$ |
| | | Supervised Methods | | | | | | |
| | GCN | $\underline{0.6892}_{\pm0.004}$ | $0.5658_{\pm0.000}$ | $0.2355_{\pm0.019}$ | $\underline{0.5277}_{\pm0.002}$ | $0.7503_{\pm0.002}$ | $0.5565_{\pm0.000}$ | $0.5542$ |
| | GAT | $0.3886_{\pm0.118}$ | $0.4997_{\pm0.012}$ | $0.3897_{\pm0.134}$ | $0.5051_{\pm0.019}$ | $0.5007_{\pm0.006}$ | $0.4977_{\pm0.006}$ | $0.4636$ |
| | BWGNN | $0.5441_{\pm0.020}$ | $0.4026_{\pm0.028}$ | $0.4214_{\pm0.039}$ | $0.4908_{\pm0.013}$ | $\underline{0.9684}_{\pm0.005}$ | $0.5841_{\pm0.062}$ | $0.5686$ |
| | GHRN | $0.5242_{\pm0.013}$ | $0.4096_{\pm0.021}$ | $0.4783_{\pm0.021}$ | $0.5036_{\pm0.016}$ | $0.9601_{\pm0.018}$ | $\mathbf{0.6045}_{\pm0.022}$ | $0.5800$ |
| | XGBGraph | $0.4869_{\pm0.069}$ | $0.4869_{\pm0.069}$ | $0.7843_{\pm0.090}$ | $0.4773_{\pm0.022}$ | $\mathbf{0.9815}_{\pm0.000}$ | $0.5869_{\pm0.014}$ | $\underline{0.6340}$ |
| | Our | $\mathbf{0.7917}_{\pm0.021}$ | $0.5356_{\pm0.005}$ | $\mathbf{0.8192}_{\pm0.015}$ | $\mathbf{0.5362}_{\pm0.007}$ | $0.9289_{\pm0.007}$ | $0.5448_{\pm0.009}$ | $\mathbf{0.6927}$ |
| AUPRC | | Unsupervised Methods | | | | | | |
| | AnomalyDAE | $\underline{0.0675}_{\pm0.028}$ | $0.0413_{\pm0.005}$ | $\mathbf{0.6172}_{\pm0.015}$ | $\underline{0.0647}_{\pm0.016}$ | $0.1025_{\pm0.026}$ | $0.1479_{\pm0.006}$ | $0.1735$ |
| | CoLA | $0.0468_{\pm0.026}$ | $0.0327_{\pm0.002}$ | $0.0956_{\pm0.005}$ | $0.0361_{\pm0.001}$ | $0.0678_{\pm0.005}$ | $0.1474_{\pm0.001}$ | $0.0711$ |
| | HCM-A | $0.0249_{\pm0.003}$ | $0.0374_{\pm0.008}$ | $0.0979_{\pm0.011}$ | $0.0511_{\pm0.000}$ | $0.0727_{\pm0.006}$ | $0.1453_{\pm0.000}$ | $0.0716$ |
| | GADAM | $0.0461_{\pm0.014}$ | $0.0299_{\pm0.004}$ | $0.0917_{\pm0.007}$ | $0.0428_{\pm0.002}$ | $0.0773_{\pm0.024}$ | $0.1602_{\pm0.010}$ | $0.0747$ |
| | TAM | $0.0243_{\pm0.002}$ | $\underline{0.0417}_{\pm0.001}$ | $0.1266_{\pm0.015}$ | $0.0532_{\pm0.002}$ | $0.1771_{\pm0.002}$ | $0.1271_{\pm0.001}$ | $0.0917$ |
| | | Supervised Methods | | | | | | |
| | GCN | $0.0437_{\pm0.001}$ | $\mathbf{0.0449}_{\pm0.000}$ | $0.2527_{\pm0.026}$ | $\mathbf{0.0763}_{\pm0.001}$ | $0.1738_{\pm0.002}$ | $0.1759_{\pm0.000}$ | $0.1279$ |
| | GAT | $0.0445_{\pm0.039}$ | $0.0327_{\pm0.001}$ | $0.0892_{\pm0.016}$ | $0.0595_{\pm0.003}$ | $0.0697_{\pm0.001}$ | $0.1478_{\pm0.003}$ | $0.0739$ |
| | BWGNN | $0.0289_{\pm0.003}$ | $0.0263_{\pm0.002}$ | $0.2735_{\pm0.026}$ | $0.0543_{\pm0.004}$ | $\underline{0.8406}_{\pm0.012}$ | $0.1975_{\pm0.031}$ | $0.2369$ |
| | GHRN | $0.0254_{\pm0.001}$ | $0.0265_{\pm0.002}$ | $0.3103_{\pm0.013}$ | $0.0541_{\pm0.005}$ | $0.8142_{\pm0.045}$ | $\mathbf{0.2015}_{\pm0.015}$ | $0.2387$ |
| | XGBGraph | $0.0268_{\pm0.006}$ | $0.0315_{\pm0.000}$ | $0.4116_{\pm0.040}$ | $0.0500_{\pm0.003}$ | $\mathbf{0.8673}_{\pm0.000}$ | $0.1994_{\pm0.012}$ | $\underline{0.2644}$ |
| | Our | $\mathbf{0.2291}_{\pm0.023}$ | $0.0340_{\pm0.001}$ | $\underline{0.4746}_{\pm0.033}$ | $0.0610_{\pm0.003}$ | $0.7329_{\pm0.042}$ | $0.1767_{\pm0.004}$ | $\mathbf{0.2847}$ |

# H INDUCTIVE LEARNING VS. ZERO-SHOT GENERALIST LEARNING

Despite our method and inductive graph learning (Hamilton et al., 2017; Xu et al., 2018) are both focused on evaluating the learned models on unseen graph data during inference, there are fundamental differences between our zero-shot learning and the inductive graph learning. We clarify the differences as follows:

- For inductive graph learning, the training dataset and testing dataset come from the same graph source. For example, for the graph classification task in Xu et al. (2018), 20 graphs of protein-protein interaction (PPI) datasets are used for training, and 2 other graphs are used for testing. These graphs both belong to the same protein-protein interaction graph dataset with the same attribute distribution and semantics. Therefore, the learned model can be easily generalized to the test graphs.

- In our zero-shot setting, the training dataset and testing dataset are from different domains and distributions. They differ in the dimensionality of node attributes and graph semantics. For example, as a shopping network dataset, Amazon contains the relationships between users and reviews, and the node attribute dimensionality is 25. Differently, Facebook, a social network dataset, describes relationships between users with 576-dimensional attributes. This is one fundamental difference between the inductive setting and our zero-shot setting. Moreover, our zero-shot setting requires the learned models to be directly applied to other graphs from different domains without any further tuning/training or labeled nodes of the target graphs. This requires the learned model to capture the more generalized patterns for anomaly detection based on only one training graph, resulting in a task being more challenging than the mentioned inductive learning.

- There are also studies (Ding et al., 2021a) on inductive graph anomaly detection, but the problem setting is also fundamentally different from our setting. In particular, to allow the evaluation of inductive detection, Ding et al. (2021a) samples nodes from the same graph to construct two graph datasets, with one graph used for training and another used for testing, leading to the fact that the training and test datasets are essentially from the same distribution. This is fundamentally different from our settings, where training and testing datasets are separately from highly heterogeneous distributions and domains.

Table 9: AUROC and AUPRC results on six real-world GAD datasets with the generalist model trained on Facebook. For each dataset and metric, the best performance per column is boldfaced, with the second-best underlined. "Avg" denotes the averaged performance of each method.

| Metric | Method | Dataset | | | | | | Avg. |
|--------|--------|---------|--------|--------|--------|-----------|-----------|------|
| | | Amazon | Reddit | Weibo | YelpChi | Aamzon-all | YelpChi-all | |
| AUROC | GraphSAGE | $0.4276_{\pm0.156}$ | $\underline{0.5275}_{\pm0.011}$ | $0.0975_{\pm0.002}$ | $0.4593_{\pm0.000}$ | $0.3276_{\pm0.074}$ | $0.4720_{\pm0.006}$ | 0.3853 |
| | AEGIS | $\underline{0.4664}_{\pm0.030}$ | $0.3530_{\pm0.016}$ | $\underline{0.4979}_{\pm0.048}$ | $\underline{0.5267}_{\pm0.016}$ | $\underline{0.4375}_{\pm0.149}$ | $\underline{0.5116}_{\pm0.022}$ | 0.4655 |
| | Ours | $\mathbf{0.7525}_{\pm0.016}$ | $\mathbf{0.5337}_{\pm0.002}$ | $\mathbf{0.8860}_{\pm0.007}$ | $\mathbf{0.5875}_{\pm0.016}$ | $\mathbf{0.7962}_{\pm0.022}$ | $\mathbf{0.5558}_{\pm0.012}$ | **0.6853** |
| AUPRC | GraphSAGE | $\underline{0.0601}_{\pm0.015}$ | $\mathbf{0.0361}_{\pm0.001}$ | $0.0858_{\pm0.000}$ | $0.0468_{\pm0.001}$ | $0.0493_{\pm0.006}$ | $0.1358_{\pm0.001}$ | 0.0690 |
| | AEGIS | $0.0600_{\pm0.003}$ | $0.0233_{\pm0.001}$ | $\underline{0.2158}_{\pm0.028}$ | $\underline{0.0677}_{\pm0.007}$ | $\underline{0.0928}_{\pm0.057}$ | $\underline{0.1628}_{\pm0.010}$ | 0.1037 |
| | Ours | $\mathbf{0.1602}_{\pm0.013}$ | $\underline{0.0351}_{\pm0.000}$ | $\mathbf{0.6406}_{\pm0.026}$ | $\mathbf{0.0712}_{\pm0.008}$ | $\mathbf{0.2430}_{\pm0.028}$ | $\mathbf{0.1810}_{\pm0.012}$ | **0.2219** |

Moreover, our problem setting is the same as existing work on zero-shot image anomaly detection (Jeong et al., 2023; Zhou et al., 2024), with the only difference in the data type used. Considering all these factors, we think "zero-shot" is more suitable for characterizing the nature of the problem complexity and more consistent with the terms/concepts used in the anomaly detection community.

To further demonstrate the difference between inductive learning and zero-shot generalist learning, we adopt the inductive learning methods to the zero-shot setting. Specifically, two representative inductive methods, GraphSAGE (Hamilton et al., 2017) and AEGIS (Ding et al., 2021a) are used and we follow the experimental setup in the main paper to unify the node attribute dimensionality with SVD. The results of GraphSAGE and AEGIS are reported in Table 9.

From the table, we can see whether the general inductive learning method or the inductive anomaly detection method does not achieve promising performance for the zero-shot generalist anomaly detection. This highlights the difference and incompatibility between inductive learning and the problem studied in this paper.

