# OpenReview forum: "Zero-shot Generalist Graph Anomaly Detection with Unified Neighborhood Prompts"
_ICLR.cc/2025/Conference — Submitted to ICLR 2025_

### Official Review · Reviewer_iwCA · 2024-10-16

**Soundness:** 3
**Presentation:** 2
**Contribution:** 1
**Rating:** 3
**Confidence:** 4

**Summary:**

This paper proposes a novel zero-shot generalist Graph Anomaly Detection (GAD) approach called UNPrompt, which aims to train a unified detection model. The approach achieves generalist GAD through two key modules: the first module aligns the dimensionality and semantics of node attributes across different graphs by employing coordinate-wise normalization in a projected space. The second module focuses on learning generalized neighborhood prompts that leverage the predictability of latent node attributes as an anomaly score across various datasets.

**Strengths:**

1. The experiments are detailed and demonstrate strong performance.
2. The paper is well-organized.
3. The work addresses a novel problem in a new context: zero-shot Graph Anomaly Detection (GAD).

**Weaknesses:**

1. The theoretical innovation is insufficient. The concept of Neighborhood Prompting appears to encompass an all-in-one method in [1], and the method of coordinate-wise normalization is not novel. The paper does not sufficiently address the unique challenges associated with applying these methods to GAD.
2. The consideration of graph anomalies is inadequate. The evaluation focuses primarily on graph uniformity or high node connectivity as criteria for determining anomalies, while neglecting the importance of attribute features.
3. The writing needs improvement; the paper contains too many lengthy sentences, making it difficult to read and understand.
4. No code provided

   [1] Sun, Xiangguo, et al. "All in one: Multi-task prompting for graph neural networks." *Proceedings of the 29th ACM SIGKDD Conference on Knowledge Discovery and Data Mining*. 2023.

**Questions:**

1. What are the unique challenges of prompt learning in Graph Anomaly Detection (GAD)?
2.  In Formula 2, the mean value will also be large when the difference in features between the two datasets is large. So, how can the two data be projected to the same space?
3. How are potential negative values from Formula 2 handled?
4. Formulas 4 and 5 suggest that the authors only consider graph uniformity or high node connectivity as criteria for determining anomalies. This design seems somewhat superficial, as anomalous nodes may also be related to node attributes.
5. Does Neighborhood Prompting involve learning a prompt for each node's neighbours? If the added prompts alter the neighbour features of *x*, could this affect the detection of anomalies?

---

> ### Author Response · Authors · 2024-11-22
> **Response to Reviewer iwCA (Part 1)**
>
> Thank you very much for the constructive comments and questions. We are grateful for the positive comments on our design and empirical justification. Please see our detailed one-by-one responses below.
>
> > **Weakness #1 & Question #1:** Contribution of coordinate-wise normalization and neighborhood prompting and the unique challenges of prompt learning for GAD.
>
> Please refer to our reply in **Global Response to Shared Concerns #1** in the **General Response** section above for this concern.
>
> We would also like to clarify the differences between our method and the all-in-one method in [Ref1]. First, [Ref1] aims to utilize prompt learning to reformulate various downstream tasks in line with the pre-training task so that the pre-trained model can be applied to downstream applications with efficient fine-tuning or even without any fine-tuning. By contrast, we focus on using neighborhood prompting to capture generalized normal and abnormal patterns so that the anomaly detection model trained on the source graph can be directly applied to target graphs. Second, [Ref1] learns a prompt graph consisting of prompt tokens and token structure. The prompt graph is then inserted into the input graph. Differently, we only devise learnable prompt tokens appending to the attributes of the neighboring nodes of the target nodes for learning robust and discriminative patterns. Therefore, the concept of neighborhood prompting is not encompassed in [Ref1].
>
> There are two unique challenges in prompt learning for graph anomaly detection. First, the unsupervised nature of graph anomaly detection in a target graph implies the lack of labeled data from the target graph datasets, rendering the prompt learning with multi-downstream tasks infeasible. By contrast, UNPrompt is designed to eliminate the tuning on the target graph datasets. Second, the anomalous patterns differ significantly from one graph to another, and thus, to achieve generalized anomaly detection across different graphs, it is crucial that the prompt learning requires a one-class pattern learning approach. Conventional graph prompt learning does not meet this criterion. Our latent node attribute prediction-based prompt learning instead is specifically designed for this purpose, in which strong latent node attribute predictability is expected for normal nodes.
>
> > **Weakness #2 & Question #4:** Inadequate consideration for GAD.
>
> In this paper, we utilize the predictability of latent node attributes as the anomaly criteria instead of using graph uniformity or high node connectivity for anomaly detection. Specifically, the working mechanism of the predictability of latent node attributes is that normal nodes tend to have more connections with normal nodes of similar attributes due to prevalent graph homophily relations. This results in a more homogeneous neighborhood for the normal nodes. By contrast, the presence of anomalous connections and/or attributes makes abnormal nodes deviate significantly from their neighbors. For a target node, its latent attributes (i.e., node embedding) are more predictable based on the latent attributes of its neighbors if the node is a normal node, compared to abnormal nodes. Therefore, both anomalous nodes and edges are considered in our method and the whole process of the anomaly score calculation is formulated in Eq. (3)-Eq. (5).
>
> > **Weakness #3:** Lengthy sentences.
>
> We will modify lengthy sentences to make the paper more readable and understandable in the revised version.
>
> > **Weakness #4:** Source Code.
>
> We will release the source code upon paper acceptance. Currently, to facilitate your review, we have uploaded the code to an anonymous GitHub repository at https://anonymous.4open.science/r/UNPrompt-DB13/README.md.

---

> > ### Author Response · Authors · 2024-11-22
> > **Response to Reviewer iwCA (Part 2)**
> >
> > > **Question #2 & Question #3:** Explanation on projecting two data into the same space and how are potential negative values from Formula 2 handled?
> >
> > In Eq.(2), we use coordinate-wise normalization to align the semantics and unify the distributions across graphs. This normalization is applied to each graph independently and unifies the node attribute of each graph to have mean zeros and variance ones. In this way, the semantics of different graphs are mapped into a common space, as shown in Figure 1 (a) and Appendix A. Therefore, it does not matter whether the difference in features between two datasets is large or there exist negative values in node attributes.
> >
> > > **Question #5:** Does Neighborhood Prompting involve learning a prompt for each node's neighbors?
> >
> > For a target node, we aim to modify the attribute of its neighbors via neighborhood prompts so that the neighborhood-based latent prediction of normal nodes is maximized while that of abnormal
> > nodes is minimized simultaneously. However, it does not mean that we learn a prompt for each node's neighbors as it would induce a substantial amount of storage resources and computation costs. In our paper, we propose to learn a set of vector-based prompt tokens for all nodes. The neighbors of each node are augmented by the weighted combination of the same set of these tokens, with the weights obtained from learnable linear projections.
> >
> > **Reference**
> >
> > - [Ref1] Sun, Xiangguo, et al. "All in one: Multi-task prompting for graph neural networks." Proceedings of the 29th ACM SIGKDD Conference on Knowledge Discovery and Data Mining. 2023.

---

> > > ### Comment · Reviewer_iwCA · 2024-11-26
> > >
> > > Thanks for the detailed responses. The two challenges clarified by the authors are more similar to the general challenges of graph anomaly detection, which have been covered in many existing work: unsupervised graph learning and graph domain adaptation. What I would like to ask is, what are the unique challenges prompt learning faces in solving the unsupervised anomaly detection problem? As I mentioned in Q5, I was concerned that the added prompt vectors would weaken the anomaly of the abnormal node and thus affect the result.

---

> > > > ### Author Response · Authors · 2024-11-27
> > > >
> > > > Thanks for your reading and suggestions. Please see our detailed response and clarification below:
> > > >
> > > > > **Question #1:** The unique challenges of prompt learning in unsupervised anomaly detection.
> > > >
> > > > We'd like to clarify that our graph prompt learning is designed for **zero-shot** GAD, rather than conventional unsupervised GAD, and so, the unique challenges jointly arise from (i) the nature of being **zero-shot** and (ii) the primary GAD task. The main challenge due to (i) is that **prompt tuning is allowed on the source data (i.e., Facebook in our experiments) and NOT allowed on the target data (i.e., the rest of GAD datasets in our paper) since no target data is available during prompt learning**. The key challenge due to (ii) is that we need to learn prompts that capture highly varying normal/abnormal patterns, since **the anomalous patterns differ significantly from one graph (i.e., Facebook) to another (i.e., any of the other GAD datasets)**. This leads to a very unique joint challenge under **zero-shot GAD**. We need to learn a set of prompts that capture normal and anomalous patterns on the source data and can generalize to different target datasets, without any further tuning/retraining.
> > > >
> > > > As discussed previously, current graph prompt learning typically follows the mechanism of tuning the prompts for a specific pre-trained model using annotated training data of target datasets from various downstream tasks, which is not applicable to the studied zero-shot GAD setting.
> > > >
> > > > Additionally, we show that our method UNPrompt is flexible and can be adapted to conventional unsupervised GAD. Then the unique challenges here include (i) the unavailability of labeled data (since effective prompt learning requires the tuning on fairly good annotated data) and (ii) how to synthesize the task of prompt learning and the learning of unsupervised graph anomaly measures. Again, existing graph prompt learning techniques do not apply to such settings as well since (i) only single graph dataset and no label information are available (ii) they are not related to unsupervised anomaly measures in any way.
> > > >
> > > > Please note that the unique challenges unavoidably involve challenges from the primary GAD task since we have to tackle the primary challenges of GAD, but there certainly invlove new challenges due to the utilization of prompt learning in tackling GAD. We hope the discussions above clarify the issues.
> > > >
> > > >
> > > > > **Question #2:** The effectiveness of the added prompts.
> > > >
> > > > The added prompts are enforced to enhance the discriminability of abnormal nodes from normal nodes, rather than to weaken the discriminability. As we discuss in lines 320-322, the prompts are learned to achieve the goal that the latent attributes (embeddings) of the **normal nodes** are pulled closer to that of its neighbors, while pushing the latent attributes (embeddings) of the **abnormal nodes** away from that of its neighbors. This is achieved by the proposed neighborhood-based latent attribute prediction module in UNPrompt. In other words, the normal and abnormal nodes can be not differentiable in the raw attribute space, but they become differentiable due to the added prompts.
> > > >
> > > > To evaluate the effectiveness of the proposed neighborhood prompts, we perform the experiments of UNPrompt without neighborhood prompts under both the generalist setting and the unsupervised setting.
> > > >
> > > > - **Generalist Setting.** Under the generalist setting, the results without neighborhood prompts are reported on page 9 of the main paper. For easier reference, we report the results again in the following table.
> > > >
> > > > ```
> > > > Table A1. AUROC results of the proposed method UNPrompt under generalist setting.
> > > > ```
> > > > |Method |Amazon |Reddit| Weibo |YelpChi |Aamzon-all| YelpChi-all| Avg.|
> > > > |---|---|---|---|---|---|---|---|
> > > > |UNPrompt| 0.7525| 0.5337| 0.8860 |0.5875 |0.7962 |0.5558 |0.6853|
> > > > |w/o Prompt| 0.5328| 0.5500| 0.4000| 0.4520| 0.4096| 0.4894 |0.4723|
> > > >
> > > > We can see from the results that the learning of latent node attribute prediction is ineffective for capturing generalized normal and abnormal patterns without the addition of neighborhood prompts.
> > > >
> > > > - **Unsupervised Setting.** Without loss of generality, three datasets are employed to evaluate the effectiveness of neighborhood prompts under the unsupervised setting, including Facebook, Reddit, and Amazon. The results of these datasets are reported in the following table.
> > > >
> > > > ```
> > > > Table A2. AUROC results of the proposed method UNPrompt under the unsupervised setting.
> > > > ```
> > > > |Method |Facebook |Reddit|Amazon |
> > > > |---|---|---|---|
> > > > |UNPrompt|0.9379|0.6067|0.7335|
> > > > |w/o Prompt|0.8455|0.5548|0.6149|
> > > >
> > > > The same finding can be observed in the unsupervised GAD setting, i.e., the performance without using the neighborhood prompts drops significantly. Overall, empirical results from both zero-shot and unsupervised GAD settings justify that the added prompt tokens do not weaken the abnormality of the abnormal nodes but largely enhance their discriminability to the normal nodes.

---

### Official Review · Reviewer_7LXR · 2024-10-17

**Soundness:** 2
**Presentation:** 3
**Contribution:** 2
**Rating:** 3
**Confidence:** 4

**Summary:**

The authors propose a zero-shot generalist GAD approach, UNPrompt. Specifically, they introduce two components: coordinate-wise normalization to unify node attributes across different graphs, and neighborhood prompt learning, which leverages latent node attribute predictability as an anomaly measure. Experiments show their framework can outperform baselines on datasets included in this paper.

**Strengths:**

S1. The paper is easy to follow.

S2. The authors propose a zero-shot generalist GAD approach including two main components, Coordinate-wise Normalization and neighborhood prompt learning.

S3. Experiments show their framework can outperform baselines on datasets included in this paper.

**Weaknesses:**

W1. The Coordinate-wise Normalization component is hardly considered as a contribution in this paper. As stated in the paper, the authors simply follow the previous work in the related areas to apply the Coordinate-wise Normalization, so the authors need to explain the differences between their design and framework from others. Otherwise, this component can only be considered as an implementation, not a contribution.

W2. The Latent Node Attribute Predictability is similar to Local Affinitiy in TAM. This anomalous score function shares a similar form of Local Affinitiy in TAM. Although the authors provide a comparison in Figure 3(b), they need to further explain the result. To be specific, changing the objective function in an unsupervised learning framework but not tuning hyperparameters carefully is hard to lead to optimal results. The authors need to illustrate whether they use the same hyperparameters as Latent Node Attribute Predictability for Local Affinitiy or they fine-tune optimal hyperparameters for Local Affinitiy in the comparison.
W3. The Neighborhood Prompting Learning follows the previous framework. Graph prompt learning has been utilized in several downstream applications, but the authors still follow the typical design of it. They need to further explain the differences between their design and other graph prompt learning frameworks to show their contributions. Otherwise, this key component can have the same issue as W1.

W4. The setting of the hyperparameters is not explained well. For example, for the unsupervised setting of UNPrompt, the authors use the "40th percentile of the normal scores" as the threshold to generate pseudo labels. However, the real anomalous rate is far less than this, which leads to many false pseudo-labels in this setting. They need to explain why they chose such a hyperparameter. Besides, they only provide the hyperparameter sensitivity of the number of prompts. It would be better to include the analysis of other hyperparameters.

W5. The datasets and baselines are not diverse enough. For datasets, the authors only include "social networks" and "co-review", which may share similar structural information. It would be better to include financial networks, like T-Finance[1]. For baselines, the most recent supervised work included in this paper is GHRN and unsupervised work is TAM, but XGBGraph[1] and GADAM[2] are two novel works in this area, separately. In addition, since Coordinate-wise Normalization is a very common design when compared with baselines, it would be better to include such a technique for baseline models.

**Questions:**

Q1. Can the pre-trained model use a different dataset other than Facebook to train and still get good performance?

Q2. Can the framework be used for large-scale datasets, like Elliptic[1]?

Q3. Can UNPrompt still outperform other baselines after they use the Coordinate-wise Normalization technique?


Reference:

1. Jianheng Tang, Fengrui Hua, Ziqi Gao, Peilin Zhao, Jia Li. GADBench: Revisiting and Benchmarking Supervised Graph Anomaly Detection. NeurIPS 2023.
2. Jingyan Chen, Guanghui Zhu, Chunfeng Yuan, Yihua Huang. Boosting Graph Anomaly Detection with Adaptive Message Passing. ICLR 2024.

---

> ### Author Response · Authors · 2024-11-22
> **Response to Reviewer 7LXR (Part 1)**
>
> Thank you very much for the constructive comments and questions. We are grateful for the positive comments on our design and empirical justification. Please see our detailed one-by-one responses below.
>
> > **Weakness #1 & #3**: Contribution of coordinate-wise normalization and neighborhood prompting.
>
> Please refer to our reply in **Global Response to Shared Concerns #1** in the **General Response** section above for this concern.
>
> > **Weakness #2**: More explanation on comparison to TAM.
>
> While both TAM and UNPrompt utilize the features/embeddings of local neighboring nodes to measure the normality of nodes, their working mechanisms are substantially different. The anomaly score function for TAM is $$s_i = -\frac{1}{|\mathcal{N}_i|}\sum{v_j \in \mathcal{N}_i} (sim(z_i, z_j))$$ where $\mathcal{N}_i$ denotes the neighborhoods of node $i$ and $z_i$ is the GNN-based node embedding. For UNPrompt, the score function is $$s_i = sim(z_i, \tilde{z}_i)$$ where $\tilde{z}_i$ is the aggregated neighborhood embeddings for node $i$. As indicated by the TAM score function, it explores the local affinity of one node by averaging the similarities between the node and all the neighbors in the latent space. As a result, this function relies on a clean graph structure, i.e., edges only connecting normal and similar nodes, to detect anomalous nodes. The presence of non-homophily edges, i.e., edges that connect normal and abnormal nodes, would degrade the effectiveness of TAM. Differently, for a node, we use the latent node attribution predictability to measure the normality of the node. The neighbors of the nodes are first aggregated in UNPrompt and then fed into a transformation layer to perform the prediction, which reduces the reliance on the clean graph structure.
>
> In Figure 3(b), we report the results of replacing latent attribute prediction with local affinity and the results, in which we show that the latent attribute predictability consistently and significantly outperforms the local affinity-based measure across all graphs. For this ablation study, we carefully tune the hyperparameters including the number of prompts and learning rate for the local affinity-based UNPrompt variant. The performance gap between latent attribute prediction and local affinity is attributed to the differences discussed above, rather than the training tricks.
>
> > **Weakness #4**: More analysis on hyperparameters.
>
> In the unsupervised graph anomaly detection setting, there is no label information available for the neighborhood prompt learning since we need to train and evaluate UNPrompt and other methods on the same unlabeled graph. To address this issue, we treat the nodes with normal scores greater than a threshold as normal nodes (**pseudo labeling of normal nodes**) and aim to maximize the latent attribute predictability of these nodes. In our experiments, the threshold is set to be the 40th percentile of all the normal scores due to a prior knowledge that the real anomalous rate is typically very small. We set the threshold to 40\% that is far greater than the anomalous rate in real-world datasets, mainly because we want to reduce the chance of mistakenly treating anomalous nodes as normal nodes, i.e., to obtain a set of high-confident normal nodes. Note that the proposed prompt learning only focuses on utilizing normal nodes for the unsupervised graph anomaly detection setting. Setting the threshold to a smaller percentile would increase the change that more abnormal nodes would be treated as normal nodes in our training, misleading the prompt learning in UNPrompt.
>
> Besides, we also evaluate the generalist anomaly detection with different common dimensionalities, with the dimensionality varying in [2, 4, 6, 8, 10, 12]. The results are reported in the following table.
>
> ```
> Table A1. AUPRC results of UNPrompt with different common dimensionalities with Facebook as the training graph.
> ```
> |Dimensionality |2 |4 |6 |8 |10 |12|
> |---|---|---|---|---|---|---|
> |Amazon |0.4561| 0.7350| 0.7256 |0.7525 |0.7992 |0.8132|
> |YelpChi| 0.4072 |0.5049 |0.5469 |0.5875 |0.5947 |0.6158|
>
> From the table, we can see that small dimensionality leads to poor generalist anomaly detection performance. This is attributed to the fact that much attribute information would be discarded with a small dimensionality. By increasing the common dimensionality, more attribute information is retained, leading to much better performance. More details are reported in Appendix G.1 of the revised paper.

---

> > ### Author Response · Authors · 2024-11-22
> > **Response to Reviewer 7LXR (Part 2)**
> >
> > > **Weakness #5 & Question 2:** More datasets and baselines.
> >
> > Please refer to our reply in **Global Response to Shared Concerns #2** in the **General Response** section above for this concern.
> >
> > When using Facebook as the training graph, the results of XGBGraph and GADAM are reported in the following table.
> >
> > ```
> > Tabel A2. AUROC results on six real-world GAD datasets with the models trained on Facebook only.
> > ```
> > |Method| Amazon |Reddit |Weibo |YelpChi |Aamzon-all |YelpChi-all |Avg.|
> > |---|---|---|---|---|---|---|---|
> > |GADAM |0.6646 |0.4532 |0.3652 |0.3376 |0.5959 |0.4829 |0.4832|
> > |XGBGraph |0.4179 |0.4601 |0.5373 |0.5722 |0.7950 |0.4945 |0.5462|
> > |UNPrompt |0.7525 |0.5337 |0.8860 |0.5875 |0.7962 |0.5558 |0.6853|
> >
> > We can see that the proposed method can still outperform the two methods, demonstrating the effectiveness of UNPrompt.
> >
> > > **Question #1:** The results of using other graphs as the training dataset.
> >
> > To further demonstrate the generalizability of UNPrompt, we conduct additional experiments by using Amazon as the training graph and testing the learned generalist model on the other six graphs. Note that, Facebook and Amazon are from different domains, social network and co-review network respectively. The AUROC results of all methods are reported in the following table.
> >
> > ```
> > Table A3. AUROC results on six real-world GAD datasets with the generalist model trained on Amazon. “Avg” denotes the averaged performance of each method
> > ```
> >
> > |Method| Facebook| Reddit| Weibo| YelpChi| Aamzon-all| YelpChi-all| Avg.|
> > |---|---|---|---|---|---|---|---|
> > |AnomalyDAE| 0.6123| 0.5799| 0.7884| 0.4788| 0.6233| 0.4912| 0.5957|
> > |CoLA| 0.5427| 0.4962| 0.3987| 0.3358| 0.4751| 0.4937| 0.4570|
> > |HCM-A| 0.5044| 0.4993| 0.4937| 0.5000| 0.4785| 0.4958| 0.4953|
> > |GADAM |0.6024 |0.4720 |0.4324 |0.4299 |0.5199 |0.5289 |0.4976|
> > |TAM| 0.5496| 0.5764| 0.4876| 0.5091|0.7525| 0.4268| 0.5503|
> > |GCN| 0.6892| 0.5658| 0.2355| 0.5277| 0.7503| 0.5565| 0.5542|
> > |GAT| 0.3886| 0.4997| 0.3897| 0.5051| 0.5007| 0.4977| 0.4636|
> > |BWGNN| 0.5441| 0.4026| 0.4214| 0.4908| 0.9684| 0.5841| 0.5686|
> > |GHRN| 0.5242| 0.4096| 0.4783| 0.5036| 0.9601| 0.6045| 0.5800|
> > |XGBGraph| 0.4869 |0.4869 |0.7843 |0.4773 |0.9815 |0.5869 |0.6340|
> > |UNPrompt| 0.7917| 0.5356| 0.8192| 0.5362| 0.9289| 0.5448| 0.6927|
> >
> > Similar to the observations when taking Facebook as the training graph, UNPrompt achieves the best average performance in terms of AUROC when training on Amazon, demonstrating the generalizability and effectiveness of UNPrompt with different training graphs. Note that, the training graph Amazon and the target graph Amazon-all come from the same distribution but have different numbers of nodes and graph structures. Intuitively, all the methods should achieve promising performance on Amazon-all. However, only a few methods achieve this goal, including BWGNN, GHRN, XGBGraph, and our method. The failures of other baselines can be attributed to the more complex graph structure of Amazon-all hinders the generalizability of these methods. Moreover, compared to BWGNN, GHRN, and XGBGraph, our method performs more stably across different datasets. This demonstrates the importance of capturing intrinsic normal and abnormal patterns for graph anomaly detection. The results of AUPRC are reported in Appendix G.4 of the revised paper.

---

> > > ### Author Response · Authors · 2024-11-22
> > > **Response to Reviewer 7LXR (Part 3)**
> > >
> > > > **Question #3:** Comparisons to baselines with coordinate-wise normalization.
> > >
> > > We conduct experiments by incorporating the proposed coordinate-wise normalization into the baselines to evaluate whether the normalization could facilitate the baselines. The AUROC results of the two best unsupervised/supervised competing methods AnomalyDAE and GCN are reported in the following table. The results of more baselines are reported in Appendix G.3 of the revised paper.
> > >
> > > ```
> > > Table A4. AUROC results of two baselines with coordinate-wise normalization (CN).
> > > ```
> > > |Method |Amazon |Reddit |Weibo |YelpChi |Aamzon-all| YelpChi-all| Avg.|
> > > |---|---|---|---|---|---|---|---|
> > > |AnomalyDAE |0.5818 |0.5016 |0.7785 |0.4837 |0.7228 |0.5002 |0.5948|
> > > |+ CN |0.4359 |0.4858| 0.4526| 0.5992 |0.2833| 0.5080 |0.4608|
> > > |GCN |0.5988 |0.5645| 0.2232 |0.5366 |0.7195 |0.5486 |0.5319|
> > > |+ CN |0.5694 |0.5349| 0.0632 |0.3954 |0.6798 |0.5550| 0.4663|
> > > |UNPrompt | 0.7525 |0.5337| 0.8860 |0.5875 |0.7962 |0.5558 |0.6853|
> > >
> > > From the table, we can see that the proposed coordinate-wise normalization does not improve the baselines consistently. This can be attributed to two reasons. First, while the proposed coordinate-wise normalization unifies the semantics of different graphs into the common space, the discrimination between normal and abnormal patterns would also be compressed. This requires the generalist anomaly detector to capture the fine-grained differences between normal and abnormal patterns. Second, these baselines are not designed to capture generalized abnormality and normality across graphs, failing to capture and discriminate the generalized nuance. By contrast, we reveal that the predictability of latent node attributes can serve as a generalized anomaly measure and learn highly generalized normal and abnormal patterns via latent node attribute prediction. As a result, such a graph-agnostic anomaly measure can be well generalized across graphs.
> > >
> > > **Reference**
> > >
> > > - [Ref1] Aodong Li, Chen Qiu, Marius Kloft, Padhraic Smyth, Maja Rudolph, and Stephan Mandt. Zero-shot anomaly detection via batch normalization. In Thirty-seventh Conference on Neural Information Processing Systems, 2023.

---

> > > > ### Comment · Reviewer_7LXR · 2024-11-25
> > > >
> > > > Thanks for the detailed response. However, I still have follow-up questions. I would appreciate it if the authors could provide more experimental results.
> > > >
> > > > Q1. As claimed in the response, the authors set the threshold to 40% because they want to reduce the chance of mistakenly treating anomalous nodes as normal nodes, but it still remains unclear how the performance will change if we change the threshold within a reasonable range. For example, as shown in the paper, the anomalous ratio of each dataset is around 5%, so I want to know if the authors can provide more experiments by varying the threshold from 5% to 50%.
> > > >
> > > > Q2. Although the authors provide several new datasets as baselines, all the datasets are not within the financial area. As I mentioned in my previous comments, there are at least two datasets in the corresponding areas to show the generalization of the proposed model, i.e., T-Finance and DGraph-Fin. Besides, the ineffectiveness of unsupervised models is well-known, especially in the graph learning area, so I wonder if the authors could conduct their experiments on the largest public anomaly detection dataset, T-Social.
> > > >
> > > > Q3. Thanks for providing the results of two baselines with coordinate-wise normalization (CN). However, could you possibly provide more baselines with the CN components as their performance may increase a lot due to the technique?
> > > >
> > > > I will keep my score now but I will seriously consider increasing it if most of my concerns are addressed.

---

> ### Author Response · Authors · 2024-11-26
>
> Thanks for your detailed reading and suggestions. Please see our detailed response and clarification below:
>
> > **Question #1:** Sensitivity analysis of the threshold by ranging it from 5\% to 50\% in unsupervised GAD.
>
> To evaluate the sensitivity of the unsupervised GAD performance of our method w.r.t the threshold, we vary the threshold in the range of [5\%, 50\%] at a step size of 5\%. Without loss of generality, three datasets are employed, including Facebook, Reddit, and Amazon. The results of these datasets are reported in the following table.
>
> ```
> Table A5. AUROC results of UNPrompt with different thresholds in unsupervised GAD.
> ```
> |Datasets|5|10|15|20|25|30|35|40|45|50|
> |---|---|---|---|---|---|---|---|---|---|---|
> |Facebook|0.8859|0.9125|0.9218|0.9231|0.9210|0.9272|0.9340|0.9379|0.9369|0.9255|
> |Reddit  |0.5667|0.5754|0.5802|0.5824|0.5942|0.5871|0.5864|0.6067|0.5906|0.5828|
> |Amazon  |0.6264|0.6960|0.7152|0.7393|0.7394|0.7463|0.7462|0.7335|0.7392|0.7389|
>
> From the table, we can see that the unsupervised performance of UNPrompt can perform well and stably with a sufficiently large threshold (e.g., no less than 30\%), but it may drop significantly with a small threshold, e.g., thresholds like 5\%-15\% that are close to the ground-truth anomaly rate. This is because more abnormal nodes would have a substantially higher chance to be mistakenly treated as normal nodes with such a small threshold, which would in turn mislead the optimization and subsequently degrade the GAD performance. By contrast, increasing the threshold would lift the acceptance bar of normal nodes, allowing the optimization to focus on high-confident normal nodes and effectively mitigate the adverse effects caused by the wrongly labeled normal nodes.
>
>
> > **Question #2:** Experiments on financial datasets in generalist GAD.
>
> To further evaluate the proposed method, we conduct the experiments on the T-Finance dataset. Specifically, the generalist anomaly detection model is trained on Facebook following the experimental setup on page 8 of the main paper. Then, the learned model is directly applied to the T-Finance dataset. The AUROC results are reported in the following table.
>
> ```
> Table A6. AUROC results on the T-Finance dataset with the models trained on Facebook only.
> ```
> |Methods|T-Finance|
> |---|---|
> |AnomalyDAE |0.2324|
> |CoLA       |0.4889|
> |HCM-A      |0.4160|
> |GADAM      |0.1382|
> |TAM        |0.2990 |
> |GCN        |0.2345 |
> |GAT        |0.5072 |
> |BWGNN      |0.5457 |
> |GHRN       |0.5324 |
> |XGBGraph   |0.3402 |
> |UNPrompt   |0.2318|
>
> From the table, we can see that all the methods do not achieve promising results on T-Finance. This is attributed to the fact that T-Finance differs significantly from the other datasets used in our paper. Specifically, T-Finance is a highly dense graph with an average degree of 1,079 while the average degree of the training Facebook is 51. Therefore, the model learned on Facebook can not be effectively generalized to T-Finance due to the significant difference. Moreover, the massive neighbors would easily result in low discrimination between nodes if the message-passing mechanism of GNNs is based on the whole neighborhood. This helps explain the poor performance of AnomalyDAE, GCN, and UNPrompt. Meanwhile, GAT, BWGNN, and GHRN achieve better performance as they employ the attention mechanism and graph spectral analysis for GAD. Note that although GADAM employs adaptive message passing for GAD, it also achieves poor performance. This is mainly due to the incompatibility of abnormal patterns between Facebook and T-Finance. We leave the exploration of more effective generalist anomaly detectors on graphs with such contrast structure for future research.
>
> Please note that we are the very first work on zero-shot GAD, and UNPrompt shows promising performance across a number of popular GAD datasets from social networks, co-review networks, co-purchase networks, and payment flow networks. Such impressive zero-shot GAD performance, together with its flexibility in gaining new state-of-the-art unsupervised GAD performance, makes a significant contribution to the GAD community. The contribution should be not weakened in any way due to its less effective performance on this very unusual GAD dataset.
>
> As for the two larger datasets with millions of nodes, DGraph-Fin and T-Social, the experiments can not be conducted due to the out-of-memory issue on our Linux server. We will include these results in the final version.

---

> > ### Author Response · Authors · 2024-11-26
> >
> > > **Question #3:** Experimental results of more baselines with the normalization components.
> >
> > In Appendix G.3 of the revised paper, we further provide the results of baselines of CoLA, TAM, BWGNN, and GHRN. Specifically, the AUROC results of all the used baselines are reported in the following table.
> >
> > ```
> > Table A7. AUROC results of several baselines with coordinate-wise normalization (CN).
> > ```
> > |Method |Amazon |Reddit |Weibo |YelpChi |Aamzon-all| YelpChi-all| Avg.|
> > |---|---|---|---|---|---|---|---|
> > |AnomalyDAE |0.5818 |0.5016 |0.7785 |0.4837 |0.7228 |0.5002 |0.5948|
> > |+ CN |0.4359 |0.4858| 0.4526| 0.5992 |0.2833| 0.5080 |0.4608|
> > |CoLA |0.4580 |0.4623 |0.3924 |0.4907 |0.4091 |0.4879 |0.4501|
> > |+ CN |0.4729 |0.5299 |0.3401 |0.3640 |0.5424 |0.4882 |0.4563|
> > |TAM| 0.4720 |0.5725 |0.4867 |0.5035 |0.7543 |0.4216 |0.5351|
> > |+ CN| 0.4509 |0.5526 |0.4723 |0.5189 |0.7580 |0.4057 |0.5264|
> > |GCN |0.5988 |0.5645| 0.2232 |0.5366 |0.7195 |0.5486 |0.5319|
> > |+ CN |0.5694 |0.5349| 0.0632 |0.3954 |0.6798 |0.5550| 0.4663|
> > |BWGNN |0.4769 |0.5208 |0.4815 |0.5538 |0.3648 |0.5282| 0.4877|
> > |+ CN |0.4745 |0.4942 |0.2538 |0.4727 |0.6307 |0.5221 |0.4747|
> > |GHRN |0.4560 |0.5253 |0.5318 |0.5524 |0.3382 |0.5125| 0.4860|
> > |+ CN |0.4308 |0.5061 |0.2621 |0.4781 |0.5712 |0.5200| 0.4614|
> > |UNPrompt | 0.7525 |0.5337| 0.8860 |0.5875 |0.7962 |0.5558 |0.6853|
> >
> > From the table, we can see that while the proposed coordinate-wise normalization does significantly improve the performance of some baselines on specific datasets, such as CoLA on Amazon-all, the performance on most datasets for all the baselines drops with the normalization, resulting in inferior averaged performance. This can be attributed to the reasons discussed in our previous response above.

---

> > > ### Comment · Reviewer_7LXR · 2024-11-28
> > >
> > > Thanks for the response. However, there are still some concerns that can not be addressed by the explanations and experiments provided by the authors. To be specific, although the authors provide some intuitive explanations of the CN component, the novelty of this component is limited as the major contribution. Besides, the proposed model failed to generalize the performance to another dataset, i.e., T-Finance. At the same time, some other baselines can achieve a relatively good performance, severely degrading the convincingness of the proposed model as the structures of graphs in real deployment can vary a lot. Furthermore, The authors failed to provide the results of two commonly used graph anomaly detection datasets, DGraph-Fin and T-Social, which also raises a question about the practical values of this framework due to the ubiquitous large-scale graphs nowadays. In addition, there are some minor issues with the response. For example, in Table A5, they didn't provide the performance of 35% Facebook.
> > >
> > > In conclusion, although the authors put a lot of effort into providing more experiments, with the concerns mentioned above, I would like to keep negative but also tend to increase my score to 4, which can be considered as a weak reject.

---

> > > > ### Author Response · Authors · 2024-11-28
> > > >
> > > > Dear Reviewer 7LXR,
> > > >
> > > > Many thanks for the follow-up comment.
> > > >
> > > > Please be noted that we don't claim any technical novelty in the CN component throughout the paper and our rebuttal, so we believe that your comment "*the novelty of this component is limited as the major contribution*" should not be a concern. As highlighted in our paper and the rebuttal, our contributions lies in the pioneering exploration of the zero-shot GAD problem and the first very graph prompting  method for GAD. The CN is only counted as a contribution in that we empirically reveal its effectiveness in enabling zero-shot GAD.
> > > >
> > > > In terms of empirical performance, our method UNPrompt shows promising performance across  **eight popular real-world GAD datasets from social networks, co-review networks, co-purchase networks, and payment flow networks**. The dataset used well represent the challenges of having largely varying graph structure and attributes in realistic real-life application settings. Effective performance on such a diverse collection of datasets strongly indicates the feasibility of  our method UNPrompt.
> > > >
> > > > It is true that UNPrompt does not work well on T-Finance and we cannot provide results on the two very large datasets, but it is the very first method in this completely new, exciting research line. Importantly, we have provided comprehensive empirical results on a large collection of datasets and our method is the best method compared to current state-of-the-arts, well justifying what we have claimed in the paper. We don't really expect that the very first method on a challenging problem does not have any limitation. The limitations of the work leave great opportunities for us, as an AD/GAD community, to progressively tackle this new problem. We kindly request you to keep a more open-minded view on methods developed for new problems, and re-consider your rating.
> > > >
> > > > Best,

---

> > > > ### Author Response · Authors · 2024-11-28
> > > >
> > > > As for "*For example, in Table A5, they didn't provide the performance of 35% Facebook*", it was an editing problem. We have just updated the results in Table A5.
> > > >
> > > > Besides, as far as we understand, we have addressed your concerns in four out of five weaknesses. Please kindly advise if otherwise. Given all these factors, we'd like to kindly request you to re-consider your rating.

---

### Official Review · Reviewer_tYei · 2024-11-02

**Soundness:** 3
**Presentation:** 3
**Contribution:** 3
**Rating:** 6
**Confidence:** 4

**Summary:**

This paper investigates the zero-shot generalist graph anomaly detection (GAD) problem, aiming to train a one-for-all model that works on all potential datasets. A new method termed UNPrompt with several designs (including prompt tuning, attribute predictability, etc.) is provided. Experiments on both generalist and traditional settings are conducted to show the effectiveness of the proposed method.

**Strengths:**

1. The research problem of zero-shot generalist GAD is both challenging and intriguing, offering the potential to inspire further innovation within the GAD research community.
2. The proposed method is thoughtfully designed with a balance of sophistication and simplicity, making it accessible for future researchers to build upon.
3. Extensive experiments conducted across diverse scenarios demonstrate the practical effectiveness and versatility of the proposed approach.

**Weaknesses:**

1. The experimental details are not given clearly in the paper. Which graph datasets are used to train the model (for pre-training and/or prompt tuning)?
2. In the experiments, only 4 different datasets are used for evaluation (Amazon-all and YelpChi-all can be regarded as an extended version of the original ones). In this case, the generalized ability of the proposed method can't be fully verified. In this case, more datasets from different domains are expected in the experiments.
3. Why coordinate-wise normalization plays a critical role in the performance of UNPrompt? More explanation is expected.

**Questions:**

The authors are expected to answer the questions listed in Weaknesses. Apart from these, I have the following questions:
1. As ARC (Liu et al 2024) is another generalist GAD method, can you further discuss the difference between UNPrompt and ARC? Is that possible to compare them together (even though their settings are slightly different)?
2. The "attribute predictability" looks similar to the "local affinity" in TAM (Qiao and Pang 2023). In this case, will the performance drop significantly if we replace this design with local affinity?

---

> ### Author Response · Authors · 2024-11-22
> **Response to Reviewer tYei (Part 1)**
>
> Thank you very much for the constructive comments and questions. We are grateful for the positive comments on our design and empirical justification. Please see our detailed one-by-one responses below.
>
> > **Weakness #1**: More Experimental details.
>
> For experimental results reported in Table 1, the proposed UNPrompt and all competing methods are trained on Facebook and then directly tested on the other six datasets without involving any further training or label information of the target graphs. The training stage in UNPrompt involves pre-training and prompt learning. The Facebook dataset is used for both pre-training and prompt learning in UNPrompt but in a sequential order, as stated in Algorithm 1 in Appendix C.
>
> > **Weakness #2**: More datasets from different domains.
>
> Please refer to our reply in **Global Response to Shared Concerns#2** in the **General Response** section above for this concern.
>
> > **Weakness #3**: More explanation on coordinate-wise normalization.
>
> The proposed method aims to learn a generalist anomaly detection model based on one source graph and then directly apply the learned model to diverse target graphs from different domains and distributions. To achieve this goal, it is critical to learn a generalized anomaly measure across graphs, and we show that the predictability of latent node attributes can serve as such a generalized anomaly measure. However, the inconsistent node attribute dimensionality and semantics between the source graph and target graphs hinder the utilization of this generalized measure. These discrepancies are visualized and analyzed in Figure 1 (a) and Appendix A respectively. To tackle these issues, the proposed coordinate-wise normalization transforms the semantics of different graphs into a common space, as shown in Figure 1 (a) and Appendix A. In this way, the diverse distributions of node attributes are calibrated into the same semantic space and the anomaly measure learned on the source graph can effectively generalize to target graphs. The effectiveness of coordinate-wise normalization is also evaluated in the ablation study (Table 2) where the performance of generalist anomaly detection drops significantly without the normalization.
>
> > **Question #1**: Discuss the difference between UNPrompt and ARC.
>
> While both ARC and our UNPrompt focus on generalist graph anomaly detection, they have significant differences between them in terms of problem setting, methodology and experimental setup.
>
> - **Problem setting**: ARC addresses a few-shot graph anomaly detection setting that a small number of labeled normal nodes in each target graph is made available during inference, whereas UNPrompt addresses a zero-shot graph anomaly detection setting that does not require access to any labeled data in the target graph during inference.
>
> - **Methodology**: To align the node attributes of different graph datasets, ARC proposes to sort the attributes based on feature smoothness while we employ a simple yet efficient coordinate-wise normalization module. For anomaly measure, ARC utilizes in-context learning to calculate the anomaly score based on the reconstruction errors of query node embeddings w.r.t. in-context node embeddings. Differently, UNPrompt proposes a novel generalized anomaly measure, the predictability of latent node attributes, to support the learning of generalized prompt embeddings for generalist anomaly detection. In particular, UNPrompt learns graph-agnostic normal and abnormal patterns via a neighborhood-based latent attribute prediction task, in which we incorporate learnable prompts into the normalized attributes of the neighbors of a target node to predict the latent attributes of the target node. Therefore, ARC and UNPrompt takes significantly different methods to achieve the generalist models due to the difference in the problem setting.
>
> - **Experimental Setup**: The training datasets for ARC consist of a collection of the largest graphs from all multiple graph types/domains. The test graphs from the same types/domains as the training data are used during inference. By contrast, UNPrompt learns the generalist model solely on the Facebook dataset which is a small social network and the testing datasets contain both social networks and co-review networks, resulting in a more challenging and practical scenario for generalist graph anomaly detection.
>
> As we discussed above, ARC is a few-shot method that requires some labeled normal nodes in the target graphs to perform the detection task, which is not viable in the zero-shot setting used in UNPrompt. Therefore, it is not possible to compare these two methods together.

---

> > ### Author Response · Authors · 2024-11-22
> > **Response to Reviewer tYei (Part 2)**
> >
> > > **Question #2**: The performance of local affinity in UNPrompt.
> >
> > Despite the use of feature/embedding information from the local neighborhood, the latent attribute predictability in UNPrompt is different from the local affinity measure in [Ref1]. In the experimental section, we conduct the experiments by replacing the latent attribute predictability-based anomaly measure with the local affinity-based anomaly measure proposed in [Ref1], and the results are shown in Figure 3(b). The results show that the latent attribute predictability consistently and significantly outperforms the local affinity-based measure across all graphs, demonstrating its superiority in learning generalized patterns for generalist graph anomaly detection.
> >
> > **Reference**
> >
> > - [Ref1] Hezhe Qiao and Guansong Pang. Truncated affinity maximization: One-class homophily modeling for graph anomaly detection. Advances in Neural Information Processing Systems, 36, 2023.

---

> > > ### Comment · Reviewer_tYei · 2024-11-24
> > >
> > > I truly appreciate the detailed clarification, which has addressed most of my concerns. I will maintain my positive score.

---

> > > > ### Author Response · Authors · 2024-11-24
> > > >
> > > > We’re happy to be confirmed that your most concerns have been properly addressed and greatly appreciate your positive recommendation. Thank you very much.

---

### Official Review · Reviewer_8dLv · 2024-11-03

**Soundness:** 2
**Presentation:** 2
**Contribution:** 1
**Rating:** 3
**Confidence:** 4

**Summary:**

The paper proposes a method for Graph Anomaly Detection (GAD), which aims to identify nodes in a graph that significantly deviate from normal patterns. The GAD framework is trained on one graph and tested on other graphs, addressing a critical challenge in the detection of anomalies in graph-structured data.

**Strengths:**

1. The focus on graph anomaly detection is timely and relevant, as it addresses an important area within machine learning and data analysis.

2. The approach of training on one graph and testing on others is interesting.

**Weaknesses:**

1-The authors acknowledge in their rebuttal that the reported baseline results are suboptimal, which undermines the overall findings. They justify their choice of dimensionality by claiming it is heuristic: 'For all datasets used in our paper, the minimum dimensionality is 25 for the Amazon dataset. Therefore, any number that is smaller than 25 can be used as the common dimensionality. In our experiments, we chose a relatively medium size, eight, as the common dimensionality.'  However, this explanation is unconvincing, as the choice of eight seems arbitrary and indistinguishable from random.
It is important to note that this hyperparameter has a significant impact on the results (as mentioned in their rebuttal). Furthermore, the authors rely excessively on heuristic-based experimental settings for other baselines, which undermines the credibility of their comparative results.

2- The authors have not thoroughly reviewed the related work, which obscures the clarity of the novelty.

**Questions:**

Can the authors clarify the specific differences between the "ZERO-SHOT" methodology presented in this paper and established inductive settings in graph neural networks?

---

> ### Author Response · Authors · 2024-11-22
>
> Thank you very much for the constructive comments and questions. We are grateful for the positive comments on our design and empirical justification. Please see our detailed one-by-one responses below.
>
> > **Weakness and Question**: The difference between the zero-shot methodology and inductive settings.
>
> Despite our method and inductive graph learning are both focused on evaluating the learned models on unseen graph data during inference, there are fundamental differences between our zero-shot learning and the inductive graph learning. We clarify the differences as follows:
>
> - For inductive graph learning, the training dataset and testing dataset come from the same graph source. For example, for the graph classification task in [Ref1], 20 graphs of protein-protein interaction (PPI) datasets are used for training, and 2 other graphs are used for testing. These graphs both belong to the same protein-protein interaction graph dataset with the same attribute distribution and semantics. Therefore, the learned model can be easily generalized to the test graphs.
>
> - In our zero-shot setting, the training dataset and testing dataset are from different domains and distributions. They differ in the dimensionality of node attributes and graph semantics. For example, as a shopping network dataset, Amazon contains the relationships between users and reviews, and the node attribute dimensionality is 25. Differently, Facebook, a social network dataset, describes relationships between users with 576-dimensional attributes (please refer to the visualization of the distribution discrepancy in Appendix A). This is one fundamental difference between the inductive setting and our zero-shot setting. Moreover, our zero-shot setting requires the learned models to be directly applied to other graphs from different domains without any further tuning/training or labeled nodes of the target graphs. This requires the learned model to capture the more generalized patterns for anomaly detection based on only one training graph, resulting in a task being more challenging than the mentioned inductive learning.
>
> - There are also studies [Ref2] on inductive graph anomaly detection, but the problem setting is also fundamentally different from our setting. In particular, to allow the evaluation of inductive detection, [Ref2] samples nodes from the same graph to construct two graph datasets, with one graph used for training and another used for testing, leading to the fact that the training and test datasets are essentially from the same distribution. This is fundamentally different from our settings, where training and testing datasets are separately from highly heterogeneous distributions and domains.
>
> Moreover, our problem setting is the same as existing work on zero-shot image anomaly detection [Ref3, Ref4], with the only difference in the data type used. Considering all these factors, we think "zero-shot" is more suitable for characterizing the nature of the problem complexity and more consistent with the terms/concepts used in the anomaly detection community.
>
> **Reference**
>
> - [Ref1] Xu, Keyulu, et al. "Representation learning on graphs with jumping knowledge networks." International Conference on Machine Learning, 2018.
> - [Ref2] Ding, Kaize, et al. "Inductive anomaly detection on attributed networks." Proceedings of the twenty-ninth international conference on international joint conferences on artificial intelligence. 2021.
> - [Ref3] Jeong, Jongheon, et al. "Winclip: Zero-/few-shot anomaly classification and segmentation." Proceedings of the IEEE/CVF Conference on Computer Vision and Pattern Recognition. 2023.
> - [Ref4] Zhou, Qihang, et al. "Anomalyclip: Object-agnostic prompt learning for zero-shot anomaly detection." ICLR 2024.

---

> > ### Comment · Reviewer_8dLv · 2024-11-22
> >
> > Since both the proposed method and inductive graph learning focus on evaluating learned models on unseen graph data during inference, inductive graph learning methods should be thoroughly discussed in the paper and compared in the experiments. Without such a comparison, it remains unclear how the authors can demonstrate that existing inductive models fail to generalize to the problem addressed in this study.

---

> ### Author Response · Authors · 2024-11-23
>
> To further demonstrate the difference between inductive learning and zero-shot generalist learning, we adopt the inductive learning methods to the zero-shot setting. Specifically, two representative inductive methods, GraphSage [Ref5] and AEGIS [Ref2], are used and we follow the experimental setup in the main paper. The AUROC results of GraphSage and AEGIS are reported in the following table.
>
> ```
> Table A1. AUROC results on six real-world GAD datasets with the generalist model trained on Facebook.
> ```
> |Method| Amazon |Reddit| Weibo |YelpChi |Aamzon-all |YelpChi-all| Avg.|
> |---|---|---|---|---|---|---|---|
> |GraphSage |0.4276 |0.5275 |0.0975 |0.4593 |0.3276 |0.4720 |0.3853|
> |AEGIS |0.4664 |0.3530 |0.4979 |0.5267 |0.4375 |0.5116 |0.4655|
> |Ours |0.7525 |0.5337 |0.8860 |0.5875| 0.7962 |0.5558 |0.6853|
>
> From the table, we can see that either the general inductive graph learning method or the inductive GAD method does not achieve promising performance for the zero-shot generalist GAD task. This highlights the difference and incompatibility between inductive learning and the problem studied in this paper. The results of AUPRC are reported in Appendix H of the revised paper and we will include a subsection in related work to discuss the graph inductive learning studies and our differences to them.
>
> **Reference**
>
> - [Ref5] Hamilton, Will, Zhitao Ying, and Jure Leskovec. "Inductive representation learning on large graphs." Advances in neural information processing systems 30 (2017).

---

> > ### Comment · Reviewer_8dLv · 2024-11-24
> >
> > Thank you to the authors for their response. However, I still have some questions.
> >
> > -It remains unclear whether the authors have thoroughly reviewed papers related to the inductive setting, which is closely aligned with the topic of this paper. The comparison methods referenced, even in this rebuttal, are not recent, with publication years of 2017 and 2021. In addition to gaps in the related works, Section 3.1 (Preliminaries) also fails to correctly introduce conventional GAD.
> >
> > I would like to see a clear and comprehensive overview of the existing categories of inductive GNN/GAD, along with an explanation of why they are not applicable to this case. A more in-depth discussion would be more informative than a brief mention of one or two papers.
> >
> > -I am also unclear on how the authors implemented GraphSAGE and AEGIS in this paper. You mention that the attribute sizes differ across datasets—how did you address this when implementing GraphSAGE and AEGIS?

---

> > > ### Author Response · Authors · 2024-11-25
> > >
> > > To further clarify your concerns, we have added a subsection in the related work to provide a more comprehensive overview of recent advances in inductive graph learning. The additions are:
> > >
> > > "**Inductive Graph Learning**. Similar to generalist setting, inductive graph learning [Ref2, Ref5, Ref6, Ref7, Ref8] also focuses on inference on unseen graph data. However, these methods are not applicable to the generalist setting. Specifically, inductive graph learning trains the model on partial data of the whole graph dataset [Ref2, Ref5, Ref6] or the previously observed data of dynamic graphs [Ref8]. Then, the learned model is evaluated on the unseen data of the whole dataset or the future graph. These unseen testing data are from the same source of the training data with the same dimensionality and semantics. In contrast, the unseen data in our method are from different distributions/domains with significantly different dimensionality and semantics. This cross-dataset nature, specifically referred to as a zero-shot problem [Ref3, Ref4], makes our setting significantly different from the current inductive graph learning setting."
> > >
> > > In section 3.1, we have also modified this part to make conventional GAD clearer.
> > >
> > > When implementing the GraphSAGE and AEGIS, to address the dimensionality inconsistency of node attributes across graphs, we take exactly the same step as the experimental setup in page 8 of the main paper to unify the attributed dimensionality, i.e., we employ the SVD to project the node attributes of different graphs to the fixed dimensionality.
> > >
> > > **Reference**
> > >
> > > - [Ref6] Li, Xujia, et al. "Diga: guided diffusion model for graph recovery in anti-money laundering." Proceedings of the 29th ACM SIGKDD Conference on Knowledge Discovery and Data Mining. 2023.
> > > - [Ref7] Huang, Yihong, et al. "Unsupervised graph outlier detection: Problem revisit, new insight, and superior method." 2023 IEEE 39th International Conference on Data Engineering (ICDE). IEEE, 2023.
> > > - [Ref8] Fang, Lanting, et al. "Anonymous edge representation for inductive anomaly detection in dynamic bipartite graph." Proceedings of the VLDB Endowment 16.5 (2023): 1154-1167.

---

> > > > ### Comment · Reviewer_8dLv · 2024-11-25
> > > >
> > > > These responses still do not fully address my concerns.
> > > >
> > > > First, the authors employ SVD to project the node attributes of different graphs into an eight-dimensional space. Why was this specific dimensionality chosen? Is there evidence to support that this choice is optimal for both the proposed method and the baseline models?
> > > >
> > > > Second, many inductive learning methods are applicable to graphs across domains and typically require minimal fine-tuning. I believe these methods would provide more appropriate baseline comparisons.

---

> > > > > ### Author Response · Authors · 2024-11-26
> > > > >
> > > > > We address your concerns as follows;
> > > > >
> > > > > - When using SVD for feature projection, the number of projected dimensionality should be smaller than the number of instances and original dimensionality. For all datasets used in our paper, the minimum dimensionality is 25 for the Amazon dataset. Therefore, any number that is smaller than 25 can be used as the common dimensionality. In our experiments, we chose a relatively medium size, eight, as the common dimensionality. To further evaluate the generalist anomaly detection with different common dimensionalities, we vary the dimensionality in the range of [2, 4, 6, 8, 10, 12]. The results are reported in the following table.
> > > > >
> > > > > ```
> > > > > Table A2. AUROC results of UNPrompt with regard to different common dimensionalities with Facebook as the training graph.
> > > > > ```
> > > > > |Dimensionality	|2	|4	|6	|8	|10	|12|
> > > > > |---|---|---|---|---|---|---|
> > > > > |Amazon	|0.4561	|0.7350	|0.7256	|0.7525|0.7992|0.8132|
> > > > > |YelpChi|0.4072	|0.5049	|0.5469	|0.5875|0.5947|0.6158|
> > > > >
> > > > > From the table, we can see that small dimensionality leads to poor generalist anomaly detection performance. This is attributed to the fact that essential attribute information would be discarded with a smaller dimensionality. By increasing the common dimensionality, more attribute information is retained, leading to much better performance. More details are reported in Appendix G.1 of the revised paper.
> > > > >
> > > > > - Although there may exist inductive learning methods that can be applicable to graphs across domains with fine-tuning, these methods are not applicable to our setting due to their requirement on fine-tuning to work well. We'd like to recall that the studied zero-shot setting requires the learning of a generalist model on a training graph and directly applying the learned model to other graphs **without any re-training and fine-tuning during inference**. From another perspective, it is reasonable to argue that the strong zero-shot generalization gained by existing foundation models on different datasets and domains is due to its remarkable success in learning some sort of inductive bias, but we cannot say that conventional inductive learning techniques are supposed to also have similar zero-shot capabilities. In a similar sense, the power of conventional inductive graph learning techniques on zero-shot settings is largely restricted by their inherent designs that require fine-tuning on the target datasets to perform well. We agree that it is important to discuss the line of research on inductive graph learning and empirically compare with them, but they indeed don't work well under the zero-shot setting due to the reasons mentioned above. This is also why we need to promote a different approach, graph prompting, to achieve a strong zero-shot GAD performance.

---

> > > > > > ### Comment · Reviewer_8dLv · 2024-11-26
> > > > > >
> > > > > > Can I assume that the dimension of 8 is not the optimal hyperparameter for all baselines, but rather a randomly selected value, and that this hyperparameter dimension has not been evaluated in the other baselines?

---

> > > > > > > ### Author Response · Authors · 2024-11-26
> > > > > > >
> > > > > > > Please be advised that the setting of this dimensionality size was not specifically tuned for any methods, including our method; it was set in a heuristic way mentioned above. Hope this help clarify. Thank you for your follow-up question.

---

> > > > > > > > ### Comment · Reviewer_8dLv · 2024-11-28
> > > > > > > >
> > > > > > > > Since the authors have acknowledged that the reported baseline results are not optimal, and although they claim that the dimensionality hyperparameter selection is heuristic and acceptable, I still find the reported results unconvincing. Therefore, I will maintain my negative score.

---

> > > > > > > > > ### Author Response · Authors · 2024-11-28
> > > > > > > > >
> > > > > > > > > Thanks for the follow-up comment.
> > > > > > > > >
> > > > > > > > > This particular hyperparameter is not tuned for not only the baselines but also our method, so it is a fair comparison. Thus, the results should be convincing. Given the zero-shot nature of the studied problem, how could we tune such hyperparameters for the methods?
> > > > > > > > >
> > > > > > > > > If the reviewer still finds the results unconvincing, we would greatly appreciate if the reviewer could suggest what specific steps we could do to make the results more convincing. Thank you.

---

> > > > > > > > > ### Author Response · Authors · 2024-11-28
> > > > > > > > >
> > > > > > > > > One more thing is that the reviewer's original concern is "*If the authors can clarify that "ZERO-SHOT" and "inductive settings" are indeed different, I would be willing to raise my score*" and promised to increase the score if we address this concern. We'd like to check whether we have properly addressed this concern.
> > > > > > > > >
> > > > > > > > > The setting of the dimensionality size issue has been addressed in the post above, so we don't respond to it in this post.
> > > > > > > > >
> > > > > > > > > Overall, we'd like to summarize that we believe we have properly addressed your primary concern, and the new concern on hyperparameter setting is not an issue due to the zero-shot nature of the studied problem (i.e., to our best knowledge, there is no way to properly tune this dimensionality hyperparameter). Thus, we kindly request the reviewer to re-consider his/her rating. Thank you very much for the active engagement and fruitful feedback.

---

> > > > > > > > > > ### Comment · Reviewer_8dLv · 2024-11-28
> > > > > > > > > > **About original concern**
> > > > > > > > > >
> > > > > > > > > > This issue still hasn’t deviated from my original concern. I was hoping that you would provide a detailed introduction and comparison of the inductive methods that use fine-tuning in cross-domain settings. These methods may perform better with fine-tuning, but without fine-tuning, they should still outperform methods that aren’t designed for cross-domain use. However, since I have developed doubts about the authors' experimental results, conducting these experiments no longer seems meaningful.

---

> > > > > > > > > > > ### Author Response · Authors · 2024-11-28
> > > > > > > > > > >
> > > > > > > > > > > Hi Reviewer 8dLv, we appreciate your comments, but please bear in mind that we didn't set the dimensionality hyperparameter arbitrarily, but in a heuristic way. We think you have noted this point earlier.
> > > > > > > > > > >
> > > > > > > > > > > There are always many alternative methods in a single step. Your comments appear to suggest that we need to justify the use of all these alternative methods in each single step for both our method and competing methods to compare their optimal performance. We think it is an ideal case, but the reality is that the solution space considering all these choices in every single step is huge, especially for modern ML approaches. For example, even for the basic step like the GNN architecture, there are numerous hyperparameters to set: do we search for every possible option of the GNN architecture for each method? Do all these options really change the nature of a particular method? It is an interesting question, but should it be considered in every GNN-based paper?
> > > > > > > > > > >
> > > > > > > > > > > In our work, we focus mainly on the key steps and perform careful ablation study there. We use exactly the same settings in the other steps to ensure fair comparison.
> > > > > > > > > > >
> > > > > > > > > > > The use of SVD is just because it is a commonly used feature projection method and inspired by other similar methods mentioned in the paper.
> > > > > > > > > > >
> > > > > > > > > > > Further, it is definitely NOT true that there is a magic in the single dimensionality hyperparameter setting that turns different methods into an effective method for zero-shot GAD.
> > > > > > > > > > >
> > > > > > > > > > > Overall, we respect your personal views on doing research and on the paper, but disagree with your negative conjecture on our experiments.

---

> ### Comment · Reviewer_8dLv · 2024-11-28
>
> You haven't addressed my original concern, but I’ve uncovered even more serious issues. This makes the other questions seem less significant by comparison.
>
> First, just because the proposed method didn’t report the optimal results doesn’t mean the results are valid. The fact that no method is optimal is not a reasonable justification for comparing experimental results in a research paper.
>
> Second, you know that this parameter has the most significant impact on the results. You ran your method with this parameter and claimed that the results improve as the parameter increases. But is this true for other methods as well? Have you conducted experiments to verify this?
>
> Third, regarding your use of SVD to reduce the dimensions of other methods—do you really think this is the best approach? It might work best for your method, but is it also the best for the baselines? Have you conducted ablation studies to confirm this?
>
> Finally, even if I reluctantly accept your use of SVD, it’s unreasonable that you’ve arbitrarily chosen the parameter value. Don’t you think that’s odd? At the very least, you should run experiments with different parameter values and compare the results.

---

### Author Response · Authors · 2024-11-22
**Global Response (Part 1)**

Dear all reviewers,

Thank you very much for the time and effort in reviewing our paper, and for the constructive and positive comments. Our rebuttal consists of two parts: **Global Response** where we address the shared concerns from two or more reviewers and **Individual Response** where we provide a detailed one-by-one response to address your questions/concerns individually.

>**Global Response to Shared Concerns #1**: The contribution of coordinate-wise normalization and neighborhood prompting.

- **Coordinate-wise normalization**: Although we follow the previous work [Ref1] to adopt the coordinate-wise normalization for zero-shot anomaly detection, its studied problem is different from that of UNPrompt. Specifically, [Ref1] focuses on anomaly detection with distribution shifts that are caused by noise or time. Moreover, the training and testing datasets of [Ref1] have the same attribute dimensionality and similar semantics. By contrast, we focus on generalist anomaly detection across different distributions and domains, and the node attributes in different graphs differ significantly in node attribute dimensionality and semantics. To align the dimensionality, we devise a feature projection module. The coordinate-wise normalization further transforms the semantics of different graphs into a common space, as shown in Figure 1 (a) and Appendix A. In this way, the diverse distributions of node attributes are calibrated into the same semantic space and the anomaly measure learned on the source graph can effectively generalize to target graphs. The effectiveness of coordinate-wise normalization is also evaluated in the ablation study (Table 2) where the performance of generalist anomaly detection drops significantly without coordinate-wise normalization. We agree that this normalization is simple and has been used in other anomaly detection studies. Our contribution is to empirically verify that it can be leveraged, together with feature projection, to alleviate the distribution gap for effective zero-shot generalist graph anomaly detection.

- **Neighborhood prompting**: Conventional graph prompt learning methods instruct the pre-trained model to perform various downstream tasks and the prompts are then tuned with the labeled from the downstream tasks. As a result, they are infeasible for our setting where the target graph data is not available during training and there is no tuning/re-training on the downstream task (graph anomaly detection). However, in our UNPrompt, we incorporate graph prompts into the normalized attributes of the neighbors of a target node to predict the latent attributes of the target node, a prompt learning task that is specifically designed to learn discriminative prompt embeddings for graph anomaly detection. Further, the neighborhood prompts are solely learned on the training graph and directly applied to target graphs WITHOUT any further tuning or retaining. By learning the graph prompts in the normalized node attribute space, the generalized normal and abnormal patterns are implicitly incorporated into the prompts. In other words, for the target graph, the graph prompts adaptively enhance the latent node predictability of normal nodes and minimize that of abnormal nodes simultaneously, resulting in a capability of detecting abnormal nodes across different graphs that are not seen in pre-training or prompt learning and have varying data distributions.

>**Global Response to Shared Concerns #2**: Results on graphs from different domains.

Besides the social networks and co-review graphs, we further evaluate the performance of UNPrompt on Disney and Elliptic. These two datasets consist of co-purchase network and financial network respectively, which are different from social networks and co-review graphs. The AUROC results of all competing methods are reported in the following table. Specifically, we follow the same experimental settings in the main paper.

```
Table A1. AUROC results on two new real-world GAD datasets from other domains with the models trained on Facebook only. “Avg” denotes the averaged performance of each method.
```
|Method|Disney |Elliptic |Avg|
|---|---|---|---|
|AnomalyDAE |0.4853 |0.4197 |0.4525|
|CoLA |0.4696 |0.5572 |0.5134|
|HCM-A |0.2014 |0.2975 |0.2495|
|GADAM |0.4288 |0.3922 |0.4105|
|TAM |0.4773 |0.3282 |0.4028|
|GCN |0.5000 |0.7640 |0.6320|
|GAT |0.5175 |0.6588 |0.5882|
|BWGNN |0.6073 |0.5843 |0.5958|
|GHRN| 0.5336 |0.5400 |0.5368|
|XGBGraph| 0.6692 |0.4274 |0.5483|
|UNPrompt| 0.6412 |0.5901 |0.6157|


We can see that UNPrompt can still achieve promising performance, demonstrating the generality of UNPrompt across different graphs. XGBGraph obtains the best performance on Disney but it fails to work on Elliptic and many other datasets (see Table 1 in the main text). The results of AUPRC are reported in Appendix G.2 of the revised paper.

---

> ### Author Response · Authors · 2024-11-22
> **Global Response (Part 2)**
>
> Another significant contribution made by UNPrompt is its applicability to work as a conventional unsupervised graph anomaly detector, as shown in Table 3 in the main text. This offers the very first method that not only works effectively under the zero-shot setting but also achieves the new state-of-the-art performance under the conventional full-shot setting.
>
> As for Individual Response, we have provided a detailed one-by-one response to answer/address your questions/concerns after the post of your review.
>
> We very much hope our responses have cleared the confusion, and addressed your concerns. We're more than happy to take any further questions if otherwise. Please kindly advise!
>
> Best regards,
>
> Authors of Paper 4040
>
> **Reference**
>
> - [Ref1] Aodong Li, Chen Qiu, Marius Kloft, Padhraic Smyth, Maja Rudolph, and Stephan Mandt. Zero-shot anomaly detection via batch normalization. In Thirty-seventh Conference on Neural Information Processing Systems, 2023.

---

### Meta-Review · Area_Chair_YreD · 2024-12-16

**Metareview:**

This paper aims to solve the one-model-for-one-dataset drawback in graph anomaly detection. It devises a zero-shot generalist method in which one module aligns the dimensionality and semantics of node attributes across different graphs while another module learns generalized neighborhood prompts. The predictability of latent node attributes is used as a generalized anomaly measure, and latent node attribute prediction is used to learn the generalized normal and abnormal graph patterns   in a properly normalized node attribute space. Experimental results demonstrate the effectiveness.


After rebuttal, three out of four reviewers are still negative about this manuscript, and concern the comparative result credibility and experimental details as well as theoretical innovation.  The experimental setting and related works are also not detailed in the paper. So, more efforts could be needed.

**Additional Comments On Reviewer Discussion:**

After rebuttal, three out of four reviewers are still negative about this manuscript, and concern the comparative result credibility and experimental details as well as theoretical innovation.  The experimental setting and related works are also not detailed in the paper.

---

### Decision · Program_Chairs · 2025-01-22

Reject